# Anthropogenic aerosol forcing of the AMOC and the associated mechanisms in CMIP6 models

Taufiq Hassan[1], Robert J. Allen[1], Wei Liu[1], and Cynthia A. Randles[2]

[1]Department of Earth and Planetary Sciences, University of California Riverside, Riverside, CA, 92521 USA.
[2]ExxonMobil Research and Engineering Company, Annandale, NJ, USA

**Correspondence:** Robert J. Allen (rjallen@ucr.edu)

**Abstract.** By regulating the global transport of heat, freshwater and carbon, the Atlantic Meridional Overturning Circulation (AMOC) serves as an important component of the climate system. During the late 20th and early 21st centuries, indirect observations and models suggest a weakening of the AMOC. Direct AMOC observations also suggest a weakening during the early 21st century, but with substantial interannual variability. Long-term weakening of the AMOC has been associated with increasing greenhouse gases (GHGs), but some modeling studies suggest the build up of anthropogenic aerosols (AAs) may have offset part of the GHG-induced weakening. Here, we quantify 1900-2020 AMOC variations and assess the driving mechanisms in state-of-the-art climate models from the Coupled Model Intercomparison Project phase 6 (CMIP6). The CMIP6 all forcing (GHGs, anthropogenic and volcanic aerosols, solar variability, and land use/land change) multi-model mean shows negligible AMOC changes up to ∼1950, followed by robust AMOC strengthening during the second half of the 20th century (∼1950-1990), and weakening afterwards (1990-2020). These multi-decadal AMOC variations are related to changes in North Atlantic atmospheric circulation, including an altered sea level pressure gradient, storm track activity, surface winds and heat fluxes, which drive changes in the subpolar North Atlantic surface density flux. To further investigate these AMOC relationships, we perform a regression analysis and decompose these North Atlantic climate responses into an anthropogenic aerosol-forced component and a subsequent AMOC-related feedback. Similar to previous studies, CMIP6 GHG simulations yield robust AMOC weakening, particularly during the second half of the 20th century. Changes in natural forcings, including solar variability and volcanic aerosols, yield negligible AMOC changes. In contrast, CMIP6 AA simulations yield robust AMOC strengthening (weakening) in response to increasing (decreasing) anthropogenic aerosols. Moreover, the CMIP6 all-forcing AMOC variations and atmospheric circulation responses also occur in the CMIP6 AA simulations, which suggests these are largely driven by changes in anthropogenic aerosol emissions. More specifically, our results suggest that AMOC multi-decadal variability is initiated by North Atlantic aerosol optical thickness perturbations to net surface shortwave radiation and sea surface temperature (and hence sea surface density), which in turn affect sea level pressure gradient and surface wind−and via latent and sensible heat fluxes−sea surface density flux through its thermal component. AMOC-related feedbacks act to reinforce this aerosol-forced AMOC response, largely due to changes in sea surface salinity (and hence sea surface density), with temperature (and cloud) related feedbacks acting to mute the initial response. Although aspects of the CMIP6 all-forcing multi-model mean response resembles observations, notable differences exist. This includes CMIP6 AMOC strengthening from ∼1950-1990, when the indirect estimates suggest AMOC weakening. The CMIP6 multi-model mean also underestimates the observed increase

in North Atlantic ocean heat content. And although the CMIP6 North Atlantic atmospheric circulation responses−particularly the overall patterns−are similar to observations, the simulated responses are weaker than those observed, implying they are only partially externally forced. The possible causes of these differences include internal climate variability, observational uncertainties and model shortcomings−including excessive aerosol forcing. A handful of CMIP6 realizations yield AMOC evolution since 1900 similar to the indirect observations, implying the inferred AMOC weakening from 1950-1990 (and even from 1930-1990) may have a significant contribution from internal (i.e., unforced) climate variability. Nonetheless, CMIP6 models yield robust, externally forced AMOC changes, the bulk of which are due to anthropogenic aerosols.

## 1 Introduction

The Atlantic Meridional Overturning Circulation (AMOC) is an important component of the climate system, transporting large amounts of heat and freshwater poleward (Talley, 2008; Buckley and Marshall, 2016). The AMOC exhibits variability on a range of timescales, impacting not only surface temperature, but also precipitation and sea level in several regions (Delworth and Mann, 2000; Knight et al., 2005). The AMOC plays a central role in the climate response to anthropogenic forcing (Drijfhout et al., 2012; Winton et al., 2013; Marshall et al., 2014; Kostov et al., 2014; Marshall et al., 2015; Liu et al., 2020), and has also likely played a key role in past rapid climate change and paleoclimate shifts (Broecker, 1997).

Since April 2004, the AMOC has been directly monitored at 26.5°N by the RAPID array (McCarthy et al., 2012). The RAPID array shows a strong decline in the first part of the record and a slower increase afterwards. This record suggests an overall AMOC decline at a rate as high as 0.4 Sv yr$^{-1}$ (1 Sv = $10^6$ $m^3$ $s^{-1}$) (Smeed et al., 2014; Sévellec et al., 2017; Smeed et al., 2018; Frajka-Williams et al., 2019). The causes of this recent AMOC slow-down remain highly debated, and may be related to natural decadal variability (Zhao and Johns, 2014; Jackson et al., 2016; Yan et al., 2018). However, longer-term evidence, including sea surface temperature fingerprints and coral-based proxies, also suggest AMOC weakening−by about 0.2 Sv decade$^{-1}$ during the 20th century−as part of climate change (Rahmstorf et al., 2015; Caesar et al., 2018).

Although climate models disagree on the precise magnitude of the AMOC weakening−and differ substantially in their representation of the strength and depth of the AMOC−model simulations predict AMOC weakening in response to increasing greenhouse gases (Gregory et al., 2005; Solomon et al., 2007; Drijfhout and Hazeleger, 2007; Cheng et al., 2013; Kirtman et al., 2013; Kostov et al., 2014). By the end of the 21st century, for example, models estimate a 24-39% decline in the AMOC, with larger weakening under larger increases in future GHG emissions (Weijer et al., 2020). This has been related to reduced ocean heat loss, and secondarily through increased freshwater input at high latitudes, both of which decrease the density of sea water in the subpolar North Atlantic (i.e., the sinking region) (Thorpe et al., 2001). For example, decreased Arctic sea-ice, via positive buoyancy anomalies caused by anomalous surface heat and freshwater fluxes, may help explain weakening

of the AMOC (Sévellec et al., 2017; Liu et al., 2019). Warming of the tropical Indian Ocean−by means of atmospheric teleconnections and changes in ocean salinity and circulation−may exert a stabilizing effect on the AMOC, attenuating its recent weakening (Hu and Fedorov, 2019). Overall, our understanding of how anthropogenic perturbations impact the AMOC remains limited.

There is considerable debate on the role of anthropogenic aerosols in driving North Atlantic climate variability. One study argued anthropogenic aerosols are the dominant driver of Atlantic Multidecadal Variability (a broad term encompassing Atlantic climate variability), primarily through aerosol-cloud interactions and modification of net surface shortwave radiation (Booth et al., 2012). However, this result was based on a single climate model, the Hadley Centre Global Environmental Model version 2, Earth System configuration (HadGEM2-ES). Subsurface inconsistencies between observations and HadGEM2-ES were also noted, and long-term trends in sea surface temperatures may be too sensitive to HadGEM2-ES' aerosol loading (Zhang et al., 2013). Others have also suggested a role of anthropogenic forcing, including aerosols, in driving Atlantic Multidecadal Variability (Evan et al., 2009; Otterå et al., 2010; Chang et al., 2011; Allen et al., 2015; Murphy et al., 2017; Bellomo et al., 2018). Anthropogenic aerosols may also impact the AMOC, including strengthening the AMOC and increasing the northward cross-equatorial ocean heat transport (Delworth and Dixon, 2006; Cai et al., 2006, 2007; Cowan and Cai, 2013; Collier et al., 2013; Menary et al., 2013; Cheng et al., 2013). Menary et al. (2013)−using the same HadGEM2-ES model discussed above−finds AMOC strengthening in response to increasing anthropogenic aerosols. They argue that this is primarily driven by increased salinification of the North Atlantic subpolar gyre via increased evaporation, decreased flux of ice through the Fram Strait and increased salt advection from the subtropical Atlantic. This study, like many of the earlier studies, relies on a single climate model. Very recently, however, Menary et al. (2020) use the new Coupled Model Intercomparison Project phase 6 (CMIP6) (Eyring et al., 2016) archive to show a ∼10% AMOC strengthening from 1850-1985, which they attribute to aerosol forcing. Furthermore, Ma et al. (2020) find that the projected 21st century decline of anthropogenic aerosols in CMIP5 models induces AMOC weakening. This weakening of ocean circulation is associated with a divergence of meridional oceanic heat transport, which leads to enhanced ocean heat uptake but diminished storage of oceanic heat in the subpolar North Atlantic.

The newest generation of coupled climate and earth system models, CMIP6, represents a significant opportunity to evaluate the role of external forcing, including anthropogenic aerosols, on North Atlantic climate and the AMOC. Similar to the very recent results of Menary et al. (2020), we show that a large suite of state-of-the-art climate models simulate robust strengthening of the AMOC from ∼1950-1990, and that this response is largely driven by anthropogenic aerosols. Furthermore, CMIP6 models yield robust AMOC weakening from ∼1990-2020, with anthropogenic aerosols again playing an important role. We show that this multi-decadal AMOC variability is initiated by North Atlantic aerosol perturbations to net surface shortwave radiation and surface temperature (and hence sea surface density), which in turn affect sea level pressure gradient and surface wind−and via latent and sensible heat fluxes−sea surface density flux through its thermal component. AMOC-related feedbacks act to reinforce this aerosol-forced AMOC response, largely due to changes in sea surface salinity (and hence sea surface density), with temperature (and cloud) related feedbacks acting to mute the initial response.

## 2 Methods

### 2.1 AMOC Calculation

The AMOC is defined as the maximum stream function ($\psi$) below 500 m at 28°N in the Atlantic Ocean. It is calculated by integrating the northward sea water velocity ($vo$) with depth, $z$, along the western ($x_w$) to the eastern boundaries ($x_e$) of the Atlantic Ocean:

$$\psi(z) = \int\limits_{z}^{0} \int\limits_{x_w}^{x_e} vo(x, z') dx dz'. \tag{1}$$

The AMOC percent change is estimated from the least-squares regression slope ($r_s$) of the non-normalized AMOC time series using: $100 \times \frac{r_s \times N}{AMOC(N=1)}$, where $N$ is the number of years (e.g., 30 for 1990-2020) and $AMOC(N=1)$ is the initial AMOC strength (e.g., in 1990 for 1990-2020). The quoted AMOC percent change uncertainties are estimated as the standard error, defined as $\frac{\sigma}{\sqrt{n_m}}$, where $\sigma$ represents the standard deviation across each model mean AMOC percent change and $n_m$ is the number of models.

Following prior work, we estimate an inferred AMOC as the subpolar North Atlantic (45-60°N and 0-50°W) minus the Northern Hemisphere (0-60°N and 0-360°) surface temperature anomaly, scaled by 2.3 Sv K$^{-1}$ (Rahmstorf et al., 2015). Our inferred AMOC conclusions are qualitatively the same with alternative scalings (Caesar et al., 2018), since we apply the same scaling to both observations and CMIP6.

### 2.2 SDF Calculation

The surface density flux (SDF) indicates the loss or gain of density (buoyancy) of the ocean surface due to thermal (radiation, sensible and latent heat) and haline (sea-ice melting/freezing, brine rejection, precipitation minus evaporation) exchanges (Liu et al., 2017, 2019). An increase in subpolar North Atlantic SDF is associated with strengthening of the AMOC; a decrease in SDF is associated with weakening of the AMOC. Surface density flux is define as:

$$SDF = -\alpha \frac{SHF}{c_p} - \rho(0, SST)\beta \frac{SFWF \times SSS}{1 - SSS}, \tag{2}$$

where $c_p$, SST, and SSS are the specific heat capacity and sea surface temperature and salinity, respectively; $\alpha$ and $\beta$ are thermal expansion and haline contraction coefficients; and $\rho(0, SST)$ is the density of freshwater with a salinity of zero and the temperature of SST. SHF represents the net surface heat flux into ocean (positive downward), which is estimated as a sum of shortwave (SW) and longwave (LW) radiation, sensible (SHFLX) and latent (LHFLX) heat fluxes, and heat fluxes from sea ice melting and other minor sources. SFWF represents net surface freshwater flux into ocean (positive downward) and is estimated as precipitation + runoff + ice melting – evaporation. The first term $-\alpha \frac{SHF}{c_p}$ represents the thermal contribution (TSDF); the second term $-\rho(0, SST)\beta \frac{SFWF \times SSS}{1 - SSS}$ represents the haline contribution (HSDF) to the density flux.

## 2.3 OHC Calculation

The ocean heat content (OHC) is estimated from the ocean potential temperature for each model vertical level. It is derived by spatially integrating over the North Atlantic (0-60°N; 7.5-75°W) upper-ocean (0-700 m) (e.g., Zhang et al., 2013), and then multiplying by reference values for sea water density ($\rho$) and specific heat capacity ($C$) of 1025 kg m$^{-3}$ and 3985 J kg$^{-1}$ K$^{-1}$, respectively (Palmer and McNeall, 2014). Ocean heat content is calculated for each vertical level according to the following equation:

$$\Phi_z = \sum_{i,j} \rho C \theta_{i,j,z} V_{i,j,z}, \tag{3}$$

where $\Phi_z$ is the ocean heat content for model vertical level, $z$; $\theta$ is the potential temperature at that vertical level; $V$ is the grid cell volume; and $i, j$ are the latitudes and longitudes that cover the North Atlantic. Equation (3) is subsequently integrated throughout the upper ocean (0-700 m) to get the North Atlantic upper ocean heat content, with units of Joules.

Observed OHC data comes from NOAA National Centers for Environmental Information (NCEI). This observed upper-ocean heat content is derived from a yearly averaged dataset of objectively analyzed ocean temperature anomalies since 1955 (Levitus et al., 2012).

## 2.4 Regression Analysis

We decompose the 1940-2020 North Atlantic climate response into an anthropogenic aerosol-forced component and a subsequent AMOC-related feedback. The forced response is obtained by regressing the negative of the net downward subpolar North Atlantic surface shortwave radiation time series (-1xSW), which is a proxy for changes in anthropogenic aerosols, onto various fields including for example sea surface temperature (SST), surface wind speed (SFWD), sea level pressure (PSL) etc. Spatially dependent regression coefficients (e.g., $\frac{\delta SST}{\delta(-1 \times SW)}$), or sensitivities, are based on a linear least-squares regression analysis applied to the CMIP6 ensemble mean annual mean. To isolate the AMOC-related feedback, we first remove variability associated with the aforementioned forced response. This is accomplished by multiplying the sensitivity (for a given field) by the negative of the net downward subpolar North Atlantic surface shortwave radiation time series, and subtracting this quantity from each field. We then perform a subsequent regression, where the AMOC time series is regressed onto this new field. To convert this feedback field to the same units as the forcing field, we multiply the feedback field by the regression slope between the AMOC time series and the negative of the net downward subpolar North Atlantic surface shortwave radiation time series. This latter regression slope has a value of 0.32 $\frac{Sv}{W\,m^{-2}}$ and is significant at the 95% confidence level. Similar results are obtained if the subpolar North Atlantic aerosol optical thickness time series is used (as opposed to -1xSW, not shown).

## 2.5 Decomposition of seawater density

The seawater density ($\rho$) is diagnosed offline using the CMIP6 models simulated temperature, salinity and pressure (Fofonoff and Millard Jr, 1983). The same algorithm is used to calculate both surface and sub-surface seawater density. The density trend

is decomposed into thermal and haline components according to:

$$\delta\rho = (\overline{\frac{d\rho}{dT}})\delta T + (\overline{\frac{d\rho}{dS}})\delta S, \tag{4}$$

where $\delta$ represents the trend and $\overline{\frac{d\rho}{dT}}$ and $\overline{\frac{d\rho}{dS}}$ represents the climatological partial derivative of temperature and salinity with respect to density at each grid box. For the time series, $\delta$ represents the anomaly and $\overline{\frac{d\rho}{dT}}$ and $\overline{\frac{d\rho}{dS}}$ represents the 1900-2020 annual area mean over the subpolar North Atlantic. $T$ is temperature in °C and $S$ is salinity in PSU. $\frac{d\rho}{dT}$ is the temperature derivative of density and $\frac{d\rho}{dS}$ is the salinity derivative of the density. The derivatives are calculated based on the formulas from Unesco's joint panel on oceanographic tables and standards. The first term on the right hand side of Eq. (4) represents the thermal component and the second term on the right hand side of Eq. (4) represents the haline component of $\delta\rho$.

## 2.6 Decomposition of Latent and Sensible Heat Fluxes

Using Monin-Obukhov similarity theory (Monin and Obukhov, 1954), latent (LHFLX) and sensible (SHFLX) heat fluxes can be decomposed into wind, moisture and temperature components according to:

$$LHFLX = -L_v \rho_{air} u_* q_* \tag{5}$$

$$SHFLX = -c_{p,air} \rho_{air} u_* \theta_*, \tag{6}$$

where $L_v$ is the latent heat of vaporization; $c_{p,air}$ is the specific heat capacity of air at constant pressure; $\rho_{air}$ is the surface air density; $u_*$ is the surface velocity scale (m s$^{-1}$, also referred to as the surface friction velocity); $q_*$ is the surface humidity scale (kg kg$^{-1}$); and $\theta_*$ is the surface temperature scale (K) (Grachev and Fairall, 1997; Maronga, 2014). The velocity scale can be estimated from observed or simulated surface wind stress ($\tau$) as $u_* = \sqrt{\frac{|\tau|}{\rho_{air}}}$. Given values for latent and sensible heat fluxes and Eqs. (7-8), the moisture and temperature scales can be calculated as the residual. The validity of this methodology has been verified in MERRA2, where all fields (e.g., $u_*$, $q_*$, $\theta_*$, and the surface heat fluxes) are archived.

LHFLX and SHFLX trends can then be decomposed into wind, moisture and temperature components according to:

$$\delta LHFLX \approx -L_v \rho_{air}(\overline{u_*}\delta q_* + \overline{q_*}\delta u_*) \tag{7}$$

$$\delta SHFLX \approx -c_{p,air}\rho_{air}(\overline{u_*}\delta\theta_* + \overline{\theta_*}\delta u_*) \tag{8}$$

where $\delta$ represents the trend and $\overline{u_*}$, $\overline{q_*}$ and $\overline{\theta_*}$ represent climatological values at each grid box. $\rho_{air}$ is assumed to be constant for each grid box. Cross checking the estimated and actual LHFLX and SHFLX trends shows very close agreement. The first (second) term in Eq. (9) represents the moisture (wind) component of $\delta$LHFLX. Similarly, the first (second) term in Eq. (10) represents the temperature (wind) component of $\delta$SHFLX.

## 2.7 Storm Track Activity

We define the extratropical cyclone (storm track) activity using temporal variance statistics, band-pass filtered using a 24-hour difference filer (Chang et al., 2015; Allen and Luptowitz, 2017):

$$pp = \overline{[PSL(t+24hour) - PSL(t)]^2}, \tag{9}$$

where PSL is the daily sea level pressure and $pp$ is the 24-hour difference filtered variance of sea level pressure. The overbar corresponds to time averaging over each year.

## 2.8 Anthropogenic Aerosol Effective Radiative Forcing

Anthropogenic aerosol Effective Radiative Forcing (ERF) is estimated from the net top-of-the-atmosphere (TOA) radiative fluxes (the sum of net longwave and shortwave fluxes) using $\sim$30 years of data from fixed sea surface temperature (SST) simulations (Forster et al., 2016). More specifically, anthropogenic aerosol ERF is the net TOA radiative flux difference between piClim-Control and piClim-aer simulations (i.e., piClim-aer−piClim-Control). These two simulations are identical in all ways except piClim-Control features preindustrial aerosol and precursor gas emissions whereas piClim-aer features present-day (i.e., 2014) aerosol and precursor gas emissions. Twelve models are available for the aerosol ERF calculation. The transient anthropogenic aerosol ERF is calculated in a similar fashion, using the histSST and histSST-piAer experiments. Only three models are available for the transient aerosol ERF calculation, including MIROC6, UKESM1-0-LL, and NorESM2-LM, and these simulations end in 2014.

## 2.9 Trend and Correlation Significance

Multi-model ensemble mean trends are based on the ensemble mean time series for each model. All time series are normalized by subtracting each model's long-term (1900-2020) climatology. Trends are based on a least-squares regression and significance is based on a standard $t$-test. The lead-lag correlation analysis is based on Pearson's correlation coefficient. The 95% confidence intervals for the lead-lag correlations are estimated by first transforming the Pearson's correlation coefficient ($r$) to a Fisher's z-score ($r_z$). The corresponding standard error of the $z$ distribution is defined as: $\sigma_z = \frac{1}{\sqrt{N-3}}$, where N is the number of years. The confidence interval under the transformed system is calculated as: $r_z \pm z_{\frac{\alpha}{2}} \times \sigma_z$, where $z_{\frac{\alpha}{2}}$ is calculated from the inverse of the cumulative distribution function and $\alpha$ is 0.05 for a 95% confidence interval. The transformation is reversed to obtain the lower and upper bounds of the confidence interval. Similar lead-lag correlation results are obtained under detrended and non-detrended time series.

## 3 Results

### 3.1 CMIP6 All Forcing Simulations

#### 3.1.1 Time Series

Figure 1a shows the 1900-2020 CMIP6 all forcing ensemble mean normalized AMOC time series based on 24 models and 95 realizations (Supplementary Figure 1 shows the models and number of realizations used). Relatively small change occurs up to ~1950, after which the AMOC strengthens through ~1990, and then rapidly weakens through present-day (2020). 83% (92%) of the models yield a positive (negative) AMOC trend from 1950-1990 (1990-2020). The 1950-1990 (1990-2020) ensemble mean strengthening (weakening) represents a 7.7±1.6 (−11.4±1.8) percent change (Supplementary Figure 1). These and all subsequent percent changes are relative to the beginning year of the time period (Methods Section). As these multi-decadal AMOC variations are based on the ensemble mean from a relatively large number of models, they are not due to internal climate variability. Instead, they are driven by external forcing.

We note that the non-normalized climatological AMOC strength varies considerably across CMIP6 models (Supplementary Table 1). Over the present-day (2005-2018), the CMIP6 simulated AMOC ranges from 9.1 Sv (NESM3) to 30.3 Sv (NorESM2-MM). The corresponding multi-model mean AMOC strength and one-sigma uncertainty across models is 19.8 and 5.6 Sv, respectively (similar values are obtained over the entire 1900-2020 time period at 20.5 an 5.8 Sv). This is similar to but somewhat larger than that from the RAPID array at 17.5 Sv with an interannual standard deviation of 1.4 Sv. Re-estimating Fig. 1a using only those models that simulate a climatological AMOC strength within one standard deviation of the RAPID observations (i.e., 16.1 to 18.9 Sv) yields 8 models. This model subset yields similar 1950-2020 results, including AMOC strengthening from ~1950-1990, followed by weakening (not shown).

The AMOC is related to surface density fluxes in the subpolar North Atlantic (Liu et al., 2019, 2017), which modulate deepwater formation in the deep convection region. We define the subpolar North Atlantic region as 45-60°N and 0-50°W. We get similar results with alternate definitions of the subpolar North Atlantic region (e.g. 45-65°N and 10-60°W). Figure 1 also includes the corresponding time series for the subpolar North Atlantic surface density flux (SDF), thermal (TSDF) and haline (HSDF) components. The AMOC and SDF exhibit similar multi-decadal variations, including an increase (decrease) from ~1950-1990 (1990-2020). Moreover, most of the temporal variation in SDF is consistent with TSDF. The haline SDF component (HSDF) is an order of magnitude weaker (Fig 1l). Multi-decadal variations in TSDF are largely consistent with latent (LHFLX) and sensible (SHFLX) heat fluxes (Fig. 1d-e). Similar temporal evolution also occurs for the subpolar North Atlantic surface wind (SFWD), which is a component of both LHFLX and SHFLX. Moreover, the sea level pressure gradient (dPSL) between Europe (30-45°N and 0-30°E) and the subpolar North Atlantic also exhibits similar temporal evolution consistent with surface wind variations (Fig. 1g-h), as does the subpolar North Atlantic extratropical cyclone (storm track) activity, March mixed layer depth (MMLD) and sea surface density (SSD) (Fig. 1i-k). We also mention here that the multi-decadal evolution of these variables is generally out of phase with the subpolar North Atlantic net downward surface shortwave radiation (SW; Fig. 1f).

### 3.1.2 Lead-Lag Correlations

Figure 2 shows subpolar North Atlantic lead-lag Pearson correlations ($r$; Methods Section) based on the CMIP6 all forcing annual mean ensemble mean. The subpolar North Atlantic 550 nm aerosol optical thickness (AOT; a measure of the extinction of radiation by aerosols) and SW exhibit the maximum correlation at $-0.89$ with zero lag (Fig. 2a). The subpolar North Atlantic net surface shortwave radiation and AMOC exhibit maximum correlation at $-0.84$, with SW leading the AMOC by $\sim$12 years (Fig. 2b). Similarly, the subpolar North Atlantic net surface shortwave radiation and surface temperature are maximally correlated at 0.90 with zero lag (Fig. 2c); and surface temperature and AMOC have maximum correlation of $-0.85$, with AMOC lagging by $\sim$12 years (Fig. 2d). Thus, the subpolar North Atlantic net surface shortwave radiation and surface temperature are temporally in sync with aerosol optical thickness, all three of which lead the AMOC by $\sim$12 years.

Figure 2e shows that the Europe-subpolar North Atlantic sea level pressure gradient and the subpolar North Atlantic surface wind have a maximum (and significant) correlation of 0.69 at zero lag, which is consistent with geostrophy. Similarly, maximum correlations at zero lag occur between the surface density flux and the subpolar North Atlantic surface wind; Europe-subpolar North Atlantic sea level pressure gradient; March mixed layer depth; and sea surface density ($r = 0.86$; 0.58; 0.85; and 0.92 respectively; Fig. 2g,i,k,m). Similar results exist between both the sea level pressure gradient and surface wind and the thermal component of the surface density flux; the thermal component of the surface density flux also shows a maximum and significant correlation at zero lag with latent ($r = 0.79$) and sensible ($r = 0.94$) heat fluxes (not shown). Thus, the Europe-subpolar North Atlantic pressure gradient, as well as the subpolar North Atlantic surface wind and surface density and heat fluxes are temporally in sync and significantly correlated. These responses are similar to, and generally consistent with, North Atlantic Oscillation (NAO)-like variability driving air-sea fluxes (Eden and Jung, 2001). However, correlations between these variables (i.e., SDF, SFWD, and dPSL) and the AMOC all have maximum (and significant) correlations at a 4-5 year lead, ranging from 0.66 to 0.78 (Fig. 2f,h,j). The 5-year lead correlation where the subpolar North Atlantic surface density flux leads and AMOC is likely related to signal propagation via Kelvin waves/boundary currents, which impact the AMOC in the lower latitudes (e.g., 28°N) (Kawase, 1987; Huang et al., 2000; Johnson and Marshall, 2002; Cessi et al., 2004; Zhang, 2010). We note that the TSDF leads the AMOC by 5 years (r = 0.80), while the HSDF lags the AMOC by 7 years (r = 0.54). Similarly, $SSD$ (Fig. 2n) and $SSD_T$ (not shown) lead the AMOC, but by 10 years (similar to SST) at r = 0.84 and 0.68, respectively; $SSD_S$ (not shown) lags AMOC by 4 years (r =0.68). Thus, both surface density flux and sea surface density decompositions suggest that the thermal component leads AMOC changes whereas the haline component lags AMOC changes. This will be elaborated upon in Section 3.1.3.

Figure 3 shows that the Europe-subpolar North Atlantic sea level pressure gradient and the subpolar North Atlantic surface wind, and surface density are significantly correlated with the net downward surface shortwave radiation and surface temperature, with the latter two variables leading by 6-8 years. For example, the maximum correlation between the subpolar North Atlantic surface temperature and the Europe-subpolar North Atlantic sea level pressure gradient is $-0.67$ at a 6-year lag (Figure 3a). Similarly, the maximum correlation between the subpolar North Atlantic net downward surface shortwave radiation and the Europe-subpolar North Atlantic sea level pressure gradient is $-0.65$ at a 6-year lag (Figure 3b). Similar, but somewhat

stronger correlations exist between the subpolar North Atlantic surface temperature/net downward surface shortwave radiation and both surface wind and surface density flux.

    To summarize these results, the subpolar North Atlantic net downward surface shortwave radiation, surface temperature and aerosol optical thickness lead the Europe-subpolar North Atlantic sea level pressure gradient and the subpolar North Atlantic surface wind, surface density and heat fluxes by 6-8 years (and the AMOC by 12 years); the Europe-subpolar North Atlantic

sea level pressure gradient and the North Atlantic surface wind and surface density and heat fluxes lead the AMOC by 4-5 years. Although a correlation analysis does not show causation, this analysis suggests that AMOC multi-decadal variability is initiated by North Atlantic aerosol optical thickness perturbations to net surface shortwave radiation and surface temperature, which subsequently affect the sea level pressure gradient and surface wind−and via latent and sensible heat fluxes−the surface density flux through its thermal component.

CMIP6 all forcing simulations show that multi-decadal variability of the subpolar North Atlantic net surface shortwave radiation and aerosol topical thickness lead the AMOC, as well as the atmospheric circulation (e.g., dPSL and SFWD) and SDF (Fig. 2-3). This suggests changes in anthropogenic aerosols are important drivers of North Atlantic atmospheric circulation and AMOC multi-decadal variability. Beginning near the middle of the 20th century and lasting for several decades, global anthropogenic and chemically reactive gas emissions grew quickly, particularly from North America and Europe (Hoesly et al.,

2018). In the later parts of the 20th century, while emissions from Asia continued to grow, European and North American sulfate emissions declined as a result of emission control policies. Supplementary Figures 2-3 show a consistent evolution of North Atlantic SW, AOT and anthropogenic aerosol effective radiative forcing (ERF; Methods Section). This includes relatively rapid increases in AOT and corresponding decreases in SW and ERF beginning in ∼1940 and lasting until ∼1980, and opposite changes afterwards (i.e., about 10 years prior to the AMOC responses; Fig. 2a-b), particularly over Europe.

**3.1.3   Regression Decomposition into Aerosol-Forced and AMOC Feedback Components**

    To further investigate these AMOC relationships, we perform a regression analysis and decompose the 1940-2020 North Atlantic climate response into an anthropogenic aerosol-forced component and a subsequent AMOC-related feedback. The negative of the net downward subpolar North Atlantic surface shortwave radiation time series (-1xSW) is used as a proxy for changes in anthropogenic aerosols (Section 2.4). Figure 4a shows the expected negative sensitivity (regression slope) between

SST and the subpolar North Atlantic -1xSW (i.e., an increase in aerosols is associated with SST cooling and vice versa). The SST field also exhibits a strong AMOC-related feedback that is of opposite sign to the aerosol-forced component in the subpolar North Atlantic (Fig. 4b). That is, AMOC strengthening (weakening) leads to subpolar North Atlantic SST warming (cooling), which is consistent with increases (decreases) in poleward ocean heat transport (not shown). This SST-AMOC feedback sensitivity is consistent with expectations, including previously identified AMOC-related SST fingerprints (Caesar

et al., 2018). In particular, Fig. 4b is consistent with the notion (if you flip the sign) that AMOC weakening is related to cooling in the subpolar North Atlantic (i.e., the "warming" hole) due to decreases in poleward ocean heat transport, as well as warming in the Gulf Stream region which is associated with a northwards and closer-to-the-shore shift of the Gulf Stream. Although

we acknowledge that this regression decomposition may not completely separate the forced signal from the feedback, Fig. 4 shows very good results that are consistent with expectations, and support this approach.

Figure 5 shows corresponding regression maps for SDF and its various components. An increase in aerosols (i.e., -1xSW) is associated with an increase in SDF (Fig. 5a) and vice versa (i.e., positive aerosol forced sensitivity). This is largely related to the thermal SDF component (TSDF; Fig. 5b), with smaller contributions from the haline SDF component (HSDF; Fig. 6c). In each case, the AMOC feedback (Fig. 5 c-f) acts to reinforce these aerosol forced SDF changes, with TSDF again dominating over HSDF.

The three dominant components of TSDF, including surface shortwave radiation (SWSDF), latent heat (LSDF) and sensible heat (SSDF) all contribute to its positive aerosol forced sensitivity (Fig. 5 g-i). A positive sensitivity for SWSDF is expected, as aerosols decrease surface shortwave radiation which would act to increase SDF. We suggest the positive aerosol-forced sensitivities for LSDF and SSDF are related to aerosol-induced changes in surface winds, and in particular, an aerosol-induced increase in subpolar North Atlantic surface winds which enhances LSDF and HSDF (as previously suggested). We acknowl-

edge that the aerosol-forced LSDF sensitivities exhibit a complex spatial pattern, including opposite signed regressions outside the subpolar North Atlantic. Thus, increases in aerosols are associated with increases in subpolar North Atlantic SWSDF, LSDF and SSDF that all contribute to the increase in TSDF (and vice versa). We note that in the case of HSDF (although of secondary importance to TSDF), evaporation is the most important component for both the aerosol-forced sensitivities as well as the AMOC feedback (not shown).

Figure 5j-l shows that the corresponding AMOC feedback in the subpolar North Atlantic has opposite sign for SWSDF but similar sign for LSDF and SSDF (the LSDF feedback sensitivity is also larger in magnitude than the forced sensitivity). Thus, the AMOC feedback acts to oppose the aerosol forced TSDF change through SWSDF (as will be discussed below, this is related to clouds, which in turn is related to SSTs). Aerosol forced TSDF changes, however, are reinforced through LSDF and HSDF AMOC-related feedbacks, which in turn are likely associated with the aforementioned positive SST AMOC feedback–that

is, an increase in the AMOC is associated with subpolar North Atlantic SST warming, which would be expected to increase sensible heat flux, as well as latent heat flux as moisture is tied to temperature. These latter two feedbacks dominate, since the AMOC feedback for TSDF exhibits positive sensitivity.

Figure 6 shows a similar regression analysis for several additional climate variables. The aerosol-forced surface wind (SFWD; Fig 6a) sensitivity shows positive values in the subpolar North Atlantic. This is consistent with the corresponding

sea level pressure (PSL; Fig. 6b) sensitivity, which shows negative values near the Icelandic low and positive values over the continents, including Europe (i.e., strengthened pressure gradient). This continental PSL regression is consistent with decreases (increases) in net surface shortwave radiation enhancing surface cooling (warming), leading to an increase (decrease) in sea level pressure over the continents. This altered pressure gradient, in turn, impacts the subpolar North Atlantic surface winds. Thus, an increase in aerosols is associated with strengthening of the North Atlantic sea level pressure gradient, which

in turn leads to an increase in surface winds (and vice versa). Furthermore, in both cases, the AMOC feedback (Fig. 6d-e) exhibits weaker sensitivities as compared to the aerosol-forced sensitivities. This implies the PSL-SFWD response is largely aerosol-forced, and not a feedback related to the AMOC.

Figure 6c shows corresponding regressions for total cloud cover (CLT). The aerosol forced response exhibits large positive sensitivities over most of the North Atlantic and surrounding continents, which is consistent with aerosols leading to an increase in cloud cover. This aerosol-forced increase in subpolar North Atlantic CLT would act to enhance SST cooling, and hence, AMOC strengthening via enhanced sea surface density (discussed next). Interestingly, the AMOC feedback (Fig. 6f) shows opposite signed sensitivities in the subpolar North Atlantic. This is likely related to the AMOC feedback on SST (positive sensitivities; Fig. 4b), which impacts lower-tropospheric stability and in turn, low clouds (Klein and Hartmann, 1993). In other words, when the AMOC strengthens, the subpolar North Atlantic SST warms (e.g., due to enhanced poleward ocean heat transport), which reduces lower tropospheric stability and likely low cloud cover. In contrast, AMOC weakening and the associated subpolar North Atlantic SST cooling may increase lower tropospheric stability and in turn, low clouds. As discussed above (Fig. 5), this negative CLT-AMOC feedback weakens the positive TSDF-AMOC feedback via SWSDF (but this negative feedback is not large enough to offset the LSDF and SSDF AMOC feedbacks on TSDF).

Finally, we perform a similar regression analysis on sea surface density ($SSD$), as well as its thermal ($SSD_T$) and haline ($SSD_S$) components (see Methods Section 2.5). As expected, a positive aerosol-forced sensitivity exists for $SSD$ (Fig. 7a). This is largely consistent with $SSD_T$ (Fig. 7b) as opposed to $SSD_S$ (Fig. 7c), although $SSD_S$ also contributes near the eastern boundary of the North Atlantic. Averaged over the subpolar North Atlantic, $SSD_T$ yields an aerosol-forced sensitivity of 0.042 $\frac{kg/m^3}{W/m^2}$, whereas $SSD_S$ yields a corresponding sensitivity of 0.007 $\frac{kg/m^3}{W/m^2}$ (the sum of these two yield 0.049 $\frac{kg/m^3}{W/m^2}$, which is similar to but not exactly the same as the overall SSD sensitivity of 0.044 $\frac{kg/m^3}{W/m^2}$). We note that the relative importance of salinity to the aerosol-forced $SSD$ regression (especially along the eastern boundary of the North Atlantic) is more important than the haline component was for the corresponding SDF regression (Fig. 5c).

The AMOC feedback shows similar (but weaker) positive sensitivities for $SSD$ (Fig. 7d), particularly along the eastern boundary of the North Atlantic, and this is consistent with $SSD_S$ (Fig 7f). Averaged over the subpolar North Atlantic, $SSD$ yields an AMOC feedback sensitivity of 0.006 $\frac{kg/m^3}{W/m^2}$, which is entirely due to the $SSD_S$ feedback sensitivity of 0.012 $\frac{kg/m^3}{W/m^2}$. The $SSD_T$ feedback sensitivity is of opposite sign, with a subpolar North Atlantic sensitivity of -0.008 $\frac{kg/m^3}{W/m^2}$, implying the temperature component of SSD acts to weaken the overall $SSD$ AMOC feedback. This is consistent with the previously discussed AMOC feedback on SST (positive sensitivities; Fig. 4b). Thus, the AMOC feedback acts to strengthen the $SSD$ response to aerosols, and this feedback is largely due to salinity. Moreover, this salinity AMOC feedback is larger in magnitude than the aerosol-forced salinity sensitivities (0.012 versus 0.007 $\frac{kg/m^3}{W/m^2}$, respectively). This AMOC-salinity feedback is, to some extent, consistent with recent studies that have associated AMOC weakening to reduced salinity divergence and an increase in salinity in the subtropical Atlantic (Zhu and Liu, 2020). More detailed analysis of how the salinity changes occur are warranted. We note that no clear signals are found in the poleward ocean salt transport in the Atlantic integrated over depth (in contrast to the corresponding poleward heat transport), as individual models yield contrasting responses (not shown). As salinity appears to be of secondary importance, particularly with respect to the aerosol-forced component, we defer this additional analysis to subsequent work.

In addition to the surface seawater density, we perform a similar regression analysis on subsurface seawater density ($SD$) and its thermal ($SD_T$) and haline ($SD_S$) components. For this analysis, a subset of 6 CMIP6 ALL models are used. $SD$,

$SD_T$, and $SD_S$ averaged over both 0-200 m and 0-1000 m show similar aerosol-forced and AMOC feedback sensitivities as in $SSD$, $SSD_T$, and $SSD_S$ (not shown). We also perform a regression analysis on zonal mean Atlantic $SD$, $SD_T$, and

375 $SD_S$ (Fig. 7 g-l). $SD$ yields significant and positive aerosol-forced sensitivities in depth-latitude space that are consistent with the $SSD$ results. These aerosol-forced SD sensitivities are largely consistent with $SD_T$ as opposed to $SD_S$. In terms of the corresponding AMOC feedback, $SD$ sensitivities are again significant and positive throughout most of the high-latitude North Atlantic (although weaker than the aerosol-forced sensitivities), which again supports a positive AMOC feedback. Furthermore, this feedback is largely consistent with $SD_S$ , as $SD_T$ acts in the opposite direction (i.e., negative sensitivities poleward of

380 45N). In summary, the subsurface seawater density sensitivities yield conclusions similar to those based on the sea surface density sensitivities.

  The aerosol-forced $SSD$, $SSD_T$ and $SSD_S$ results are generally consistent with those based on SDF, TSDF and HSDF (Fig. 5). However, there are some differences with the AMOC feedback, including opposite signed subpolar North Atlantic sensitivities between $SSD_T$ (negative) and TSDF (positive). The $SSD_S$ AMOC feedback is also larger than that based on

HSDF (but of the same sign). These differences are likely related to heat and salt advection, which are not directly included in the SDF calculations, but are implicitly included in $SSD$ calculations. Overall, however, we arrive at a similar picture: an increase in aerosols increases SDF and $SSD$ in the subpolar North Atlantic, and this is largely associated with the thermal component. The AMOC feedback acts to reinforce the SDF and $SSD$ aerosol-forced response. The thermal SDF AMOC feedback is most important for the SDF, with the haline SDF feedback of smaller importance. Based on $SSD$, the $SSD_T$

AMOC feedback is of the opposite sign, which acts to weaken the positive sensitivities between the AMOC and $SSD$. The $SSD_S$ feedback is most important for $SSD$, which is likely related to changes in salt advection that are not captured by HSDF. Although not shown, we also note that the aerosol-forced March mixed layer depth (MMLD) exhibits significant positive sensitivities in the subpolar North Atlantic (implying enhanced deep convection in response to aerosol forcing), and the corresponding MMLD-AMOC feedback also exhibits positive (but somewhat weaker) sensitivities, which again implies

that the AMOC induces changes that positively feedback onto the AMOC (e.g., the aforementioned salinity contribution to $SSD_S$; and the latent and sensible heat flux contributions to TSDF.

### 3.1.4 Spatial Trend Maps

Figure 8 shows the 1990-2020 CMIP6 all forcing ensemble mean annual mean spatial trend map, and the corresponding model agreement on the sign of the trend, for the surface density flux and its thermal component. Consistent with Fig. 1, SDF sig-

400 nificantly decreases from 1990-2020 in the subpolar North Atlantic, with high (80-100%) model agreement (Fig. 8a,b). Most of this SDF decrease is driven by the thermal component (Fig. 8c,d). The haline component yields a weak decrease (Supplementary Figure 4). Moreover, decomposing the thermal SDF into its respective components shows that latent and sensible heat fluxes are the dominant drivers (Supplementary Figure 4). Consistently, the CMIP6 all forcing 1990-2020 ensemble mean surface wind trend−a component of latent and sensible heat fluxes−shows significant reductions with high (80-100%) model

agreement in the subpolar North Atlantic. These surface wind trends are consistent with the corresponding PSL trends, including a weakened Europe-subpolar North Atlantic pressure gradient (Fig. 8e-h), as well as a decrease in subpolar North Atlantic

extratropical cyclone (storm track) activity (Fig. 8i-j). Near opposite changes occur from 1950-1990 (Supplementary Figure 5). March mixed layer depth (MMLD) is used to investigate North Atlantic deep convection, which is associated to deep water formation and the strength of the AMOC (Liu and Liu, 2013). Our results suggest a consistent response in March mixed layer depth (Supplementary Figure 6). 1990-2020 CMIP6 all forcing ensemble mean annual mean show a significant decrease in wintertime deep convection in the North Atlantic. During the 1950-1990 time period, the CMIP6 all forcing ensemble mean shows an increase in deep convection (but not as statistically significant as the 1990-2020 decrease). Similar trend patterns from reanalyses and observations also exist, including the subpolar North Atlantic sea level pressure, surface winds and latent and sensible heat fluxes (Supplementary Figures 7-9). However, dissimilarities in magnitude exist, suggesting these responses are only partially externally forced.

### 3.1.5 Decomposition of Latent/Sensible Heat Fluxes

Using Monin-Obukhov similarity theory (Monin and Obukhov, 1954), latent and sensible heat fluxes can be further decomposed into wind, moisture and temperature components (Methods Section). Supplementary Figures 10-11 shows the importance of wind changes to latent and sensible heat fluxes, and in turn, the thermal component of the SDF. Thus, our results suggest that strengthening (weakening) of the AMOC from ~1950-1990 (1990-2020) is in part due to strengthening (weakening) of the surface winds in the subpolar North Atlantic (consistent with the altered sea level pressure gradient), which in turn leads to increases (decreases) in surface density flux through increases (decreases) in surface latent and sensible heat fluxes.

### 3.2 CMIP6 Anthropogenic Aerosol Simulations

Figure 9a shows the 1900-2020 CMIP6 anthropogenic aerosol (AA) ensemble mean normalized AMOC time series based on 8 models and 43 realizations (Supplementary Figure 12 shows the models and number of realizations used). The evolution of the AMOC in CMIP6 AA simulations is similar to that in the corresponding all forcing simulations, in particular the strengthening from ~1950 to 1990, and weakening afterwards. 88% (100%) of the models yield a positive (negative) AMOC trend from 1950-1990 (1990-2020). The 1950-1990 (1990-2020) ensemble mean strengthening (weakening) represents a $8.8\pm2.3$ ($-7.1\pm1.6$) percent change (Supplementary Figure 12). Figure 9 also shows that from ~1950-2020, surface density and heat fluxes, as well as the sea level pressure gradient, storm track activity, and surface wind follow a similar evolution as in the CMIP6 all forcing simulations. CMIP6 AA experiments also exhibit similar lead-lag relationships as in the CMIP6 all forcing simulations (not shown).

We note that fewer CMIP6 AA (as compared to all forcing) models are available. Similar CMIP6 all forcing results as described above are generally obtained when the same 8 CMIP6 models in common between CMIP6 all forcing and CMIP6 AA are used. For example, the 1990-2020 AMOC weakening based on the original 24 all forcing models yields a percent change of $-11.4\pm1.8$. Using the 8 model subset yields similar results, at $-12.9\pm2.4$ (Supplementary Figure 13). The 8 model CMIP6 all forcing subset yields somewhat weaker 1950-1990 AMOC strengthening ($6.1\pm1.6$ versus $4.7\pm1.4$ percent change; Supplementary Figure 13).

The close correspondence between the CMIP6 AA and all forcing ensemble mean AMOC time series since ~1950 again suggests anthropogenic aerosols are driving much of the response. This is further supported by looking at the CMIP6 greenhouse gas (GHG) and natural forcing ensemble mean AMOC time series. The CMIP6 GHG ensemble mean annual mean AMOC shows long-term weakening, whereas natural forcing yields negligible long-term change (Supplementary Figures 14-15). Over 1990-2020, the CMIP6 GHG AMOC weakening represents a $-6.7\pm0.8$ percent change, which is comparable to the AMOC weakening under CMIP6 AA ($-7.1\pm1.6$; Supplementary Figure 12). Thus, ~1950-1990 AMOC strengthening in CMIP6 all forcing simulations is largely controlled by anthropogenic aerosols; from 1990-2020, both anthropogenic aerosols and GHGs contribute to AMOC weakening.

Using this common set of models, we also look at the CMIP6 Atlantic meridional streamfunction in depth-latitude space, which is calculated from zonally integrated meridional velocity field (Figure 13). The 1990-2020 CMIP6 AMOC weakening is significant throughout most of the North Atlantic in all three forcing scenarios–ALL, AA and GHG–with GHG weakening larger than that due to AA. In contrast, the 1950-1990 time period features CMIP6 ALL and AA strengthening that is again significant throughout most of the North Atlantic; CMIP6 GHG forcing yields the opposite response (and weaker than the CMIP6 AA strengthening). Thus, the 1950-1990 AMOC strengthening in CMIP6 ALL is entirely dominated by AA, with GHGs acting to mute this strengthening. The 1990-2020 AMOC weakening in CMIP6 ALL is due to both GHGs and AAs, with GHGs driving a larger response. To measure the overall 1950-2020 impact of AA versus GHGs on the AMOC, we calculate the difference of the trends (1990-2020 minus 1950-1990). Figure 13 (j-l) shows that this trend "shift" is largely due to aerosols, as opposed to GHGs.

Figure 10 shows the 1990-2020 CMIP6 AA ensemble mean annual mean trends and the model agreement on the sign of the trend for the surface density flux and its thermal component, as well as the atmospheric variables (e.g., SFWD). Responses are again very similar to the corresponding CMIP6 all forcing simulations, further supporting the importance of anthropogenic aerosols. The CMIP6 AA ensemble mean shows a decrease in SDF that is largely driven by TSDF (Fig. 10a-d), weakening of the Europe-subpolar North Atlantic pressure gradient (Fig. 10e,f), a corresponding decrease in the subpolar North Atlantic surface wind (Fig. 10g-h), a decrease in the subpolar North Atlantic storm track activity. Also consistent with CMIP6 all forcing simulations are near opposite changes in these variables from 1950-1990 (Supplementary Figure 16; see also Supplementary Figures 7-8). The CMIP6 AA ensemble mean March mixed layer depth responses are similar to the corresponding CMIP6 all forcing simulations (supplementary Figure 6), with MMLD increases (decreases) in the subpolar North Atlantic from 1950-1990 (1990-2020). And furthermore, Supplementary Figures 10-11 shows the importance of wind changes to latent and sensible heat fluxes, and in turn, the thermal component of the SDF in CMIP6 AA simulations. The AMOC strengthening in response to increasing anthropogenic aerosol forcing is consistent with prior studies (Delworth and Dixon, 2006; Cai et al., 2006, 2007; Cowan and Cai, 2013; Collier et al., 2013; Menary et al., 2013; Cheng et al., 2013). However, unlike Menary et al. (2013) who used the HadGEM2-ES model, we do not find strong evidence that increased salinification is the dominant driving factor.

## 4 Discussion

Models will continue to have uncertainties, including those relevant to the AMOC and North Atlantic climate. These include biases in the mean state, as well as their representation of the strength and depth of the AMOC (e.g., Supplementary Table 1) and ocean freshwater transport (Rahmstorf, 1996; Drijfhout et al., 2011; Danabasoglu et al., 2014; Kostov et al., 2014; Danabasoglu et al., 2016). For example, in many CMIP3/5 models, the AMOC imports freshwater into the Atlantic, in opposition to observations, likely resulting in an artificially stable AMOC (Liu et al., 2017). Models also lack realistic melting of the Greenland ice sheet and the corresponding freshening of the North Atlantic (Bakker et al., 2016).

The CMIP6 AMOC response may be too sensitive to anthropogenic aerosol forcing (e.g., Zhang et al., 2013) and CMIP6 models may also overestimate aerosol indirect effects (e.g., Toll et al., 2019). However, anthropogenic aerosol ERF estimates are consistent between CMIP6 and recent observational estimates, with 90% confidence intervals of $-1.5$ to $-0.6$ and $-2.0$ to $-0.4$ W m$^{-2}$, respectively (Bellouin et al., 2020; Smith et al., 2020). It is also notable that the aerosol ERF in CMIP5 models, with a 90% confidence interval of $-1.8$ to $-0.2$ W m$^{-2}$ (Allen, 2015), is similar to that (but with a larger range) in CMIP6 models. The mean and standard deviation of the anthropogenic aerosol ERF in 12 CMIP6 models (Supplementary Table 2) are $-0.98$ and $0.24$ W m$^{-2}$, respectively. The corresponding values in 18 CMIP5 models are $-1.0$ and $0.44$ W m$^{-2}$, respectively (Allen, 2015). In contrast, Menary et al. (2020) argues the larger 1850-1985 AMOC weakening in CMIP6 models, relative to CMIP5, is due to stronger anthropogenic aerosol forcing in CMIP6. There, they show a robust relationship between AMOC strength and a proxy for aerosol forcing−the interhemispheric difference of net top-of-the-atmosphere shortwave radiation. We note that a model's transient climate response (TCR)–the surface temperature warming around the time of $CO_2$ doubling in a 1% per year $CO_2$ increase simulation–may also be important, and CMIP6 models yield a relatively large range of 1.3 to 3.0°C (Meehl et al., 2020). Although we find the expected positive inter-model relationship between 1900-1985 AMOC changes and TCR, the correlation is only 0.28 (p = 0.18).

There is some evidence that the magnitude of the AMOC trends in CMIP6 models is related to a model's anthropogenic aerosol ERF−particularly over Europe−which again supports the importance of changes in European aerosols. The correlation (over model means and using the 12 models with aerosol ERF; Supplementary Table 2) between the global mean aerosol ERF and AMOC trend yields the expected negative (positive) correlation from 1950-1990 (1990-2020), implying models with a larger global mean aerosol ERF yield larger AMOC strengthening (weakening). However, these correlations are not significant at the 95% confidence level, at $-0.29$ for 1950-1990 and 0.11 for 1990-2020. Somewhat larger, but still non-significant, correlations between European aerosol ERF and AMOC trends exist at $-0.38$ for 1950-1990 and 0.26 for 1990-2020. Ideally, the transient aerosol ERF should be used for this calculation, but this quantity is only available for 3 models. Similar conclusions are also obtained if we divide the CMIP6 models into two groups, one with a larger (absolute value) global mean anthropogenic aerosol ERF (ERF$_{HI}$; 7 model mean aerosol ERF of $-1.17$ W m$^{-2}$), and the other with a smaller global mean aerosol ERF (ERF$_{LO}$; 5 model mean aerosol ERF of $-0.72$ W m$^{-2}$). From 1950-1990, ERF$_{HI}$ (ERF$_{LO}$) models yield AMOC strengthening that represents a $7.4\pm1.4$ ($4.7\pm2.1$) percent change. From 1990-2020, ERF$_{HI}$ (ERF$_{LO}$) models yield AMOC weakening that represents a $-14.6\pm1.6$ ($-11.3\pm2.6$) percent change (Supplementary Table 2).

The CMIP6 all forcing ensemble mean reproduces the observed Northern Hemisphere (0-60°N; 0-360°E) and the North Atlantic (0-60°N; 0-75°W) surface temperature evolution, particularly from 1950-2020 (Figure 11a,b). However, discrepancies exist in the evolution of the subpolar North Atlantic surface temperature, most notably from ~1970-1990 (Figure 11c). This may not be surprising, since this also overlaps with the 1950-1990 time period, when evolution of the CMIP6 AMOC differs from inferred (i.e., surface temperature based) AMOC observations (Methods Section). Consistently, similar discrepancies exist between inferred AMOC trends in CMIP6 and observations (Figure 11d). The CMIP6 multi-model mean shows significant 1950-1990 strengthening (0.03 Sv year$^{-1}$) whereas observations show significant weakening ($-0.03$ Sv year$^{-1}$). The sign of the inferred AMOC trend after 1990 is in better agreement between CMIP6 ($-0.07$ Sv year$^{-1}$) and observations ($-0.02$ Sv year$^{-1}$), where both show weakening. However, the observed 1990-2020 AMOC trends are weaker than the CMIP6 multi-model mean and not-significant (at the 95% confidence interval), due in part to a brief strengthening in the early to mid-1990s.

Although these CMIP6 inferred AMOC trends are comparable to the actual CMIP6 AMOC trends, there are also notable differences. The CMIP6 all forcing ensemble mean 1950-1990 inferred AMOC trend is weaker than the actual CMIP6 AMOC trend (25% weaker, 0.03 versus 0.04 Sv year$^{-1}$). And moreover, there is less model agreement for the CMIP6 1950-1990 inferred AMOC strengthening, as compared to the actual AMOC (62 versus 83%, respectively). CMIP6 1990-2020 inferred and actual AMOC trends are both significant and similar in magnitude ($-0.07$ versus $-0.08$ Sv year$^{-1}$, respectively), as is the model agreement (92% for both).

Thus, CMIP6 and observations both suggest AMOC weakening after 1990. However, disagreement exists for 1950-1990, where inferred AMOC observations show significant weakening, but CMIP6 shows significant strengthening. Moreover, disagreement exists between the CMIP6 1950-1990 actual and inferred AMOC trend, with the inferred AMOC yielding weaker and less robust strengthening. These discrepancies warrant further clarification, but they suggest that the 1950-1990 inferred AMOC in observations may yield excessive weakening (relative to the actual AMOC). A recent study suggests that the North Atlantic cooling is not only related to a weaker AMOC, but also northward heat transport. So, inferred AMOC estimates from sea surface temperature are prone to error, and they are not solely a measure of the AMOC (Keil et al., 2020). We do note, however, that multiple proxy observations, support AMOC weakening during 1950-1990 (Chen and Tung, 2018). In addition to these AMOC differences, the CMIP6 multi-model mean also underestimates the magnitude of observed increase in North Atlantic upper ocean heat content (Fig. 11e).

The inferred AMOC weakening from 1950-1990 (and even from 1930-1990) may have a significant contribution from internal (i.e., unforced) climate variability. Figure 12a shows CMIP6 AMOC trends for each individual model realization for four time periods, 1950-1990, 1990-2020, 1930-2020, as well as 1930-1990. Also included are the corresponding inferred AMOC trends based on surface temperature observations. Some individual model realizations are able to reproduce the inferred AMOC trends, including the 1950-1990 weakening, as well as weakening over the longer 1930-1990 time period. 8.6% (8 of 92) and 13% (12 of 92) of the model realizations yield 1950-1990 and 1930-1990 AMOC weakening that falls within the observational uncertainty (which includes 5 and 12 different models, respectively). For the inferred AMOC strengthening from 1990-2020, 41.3% (38 of 92) of the model realizations are within the observational uncertainty (which includes 13 models).

There are 5 realizations from two different CMIP6 models (CanESM5 and IPSL-CM6A-LR) that yield AMOC trends that fall within the observational uncertainty for all four time periods. Figure 12b shows that the corresponding ensemble mean AMOC for these 5 realizations better resembles the inferred AMOC evolution since 1900, including strengthening during the first few decades, followed by a prolonged weakening, a relatively brief strengthening, and then subsequent weakening. Furthermore, these 5 realizations also better simulate the increase in North Atlantic upper ocean heat content (Fig. 12c).
Differences remain, however, including a ∼decade delay in the initial AMOC weakening (inferred weakening begins in the 1930s but these models show weakening commences in the 1940s), as well as an earlier (and brief) strengthening during the late-20th century (inferred strengthening begins in the 1990s but these models show weakening commences in the 1980s). We note that both of these models underestimate the climatological AMOC strength relative to RAPID observations (17.5±1-standard deviation of 1.4 Sv versus 11.6 Sv for IPSL-CM6A-LR and 13.1 Sv for CanESM5; Supplementary Table 1). Although
no significant AMOC differences were found between the $ERF_{HI}$ and $ERF_{LOW}$ subsets, it is interesting to note that IPSL-CM6A-LR and CanESM5 have 2 of the lowest 5 CMIP6 anthropogenic aerosol ERFs (Supplementary Table 2). It is also possible that the reason why these two models stand out is because they have a relatively large number of realizations (11 and 10, respectively; Supplementary Figure 1), which simply increases the chances of a simulated AMOC evolution comparable to that observed.

## 5   Conclusions

CMIP6 models yield consistent multi-decadal AMOC variability, including strengthening from ∼1950-1990, followed by weakening from 1990-2020. These AMOC variations are initiated by North Atlantic aerosol optical thickness perturbations to net surface shortwave radiation and surface temperature (i.e., sea surface density), which in turn affect the sea level pressure gradient and surface wind–and via latent and sensible heat fluxes–the sea surface density flux through its thermal component.
AMOC-related feedbacks act to reinforce this aerosol-forced AMOC response, largely due to changes in sea surface salinity and its corresponding impacts on sea surface density, with temperature (and cloud) related feedbacks acting to mute the initial response. Anthropogenic aerosol forcing alone reproduces the bulk of the multi-decadal AMOC responses. Moreover, reanalyses and observations yield similar patterns of the North Atlantic atmospheric circulation response, suggesting part of this signal is externally forced. However, other aspects of the CMIP6 AMOC response are at odds with observations. This includes the
inferred ∼1950-1990 weakening of the AMOC based on surface temperature observations (e.g., Rahmstorf et al., 2015), when the CMIP6 multi-model mean yields strengthening. Moreover, the CMIP6 multi-model mean underestimates the observed increase in North Atlantic ocean heat content since ∼1955. Some of these discrepancies could be due to model shortcomings, such as excessive anthropogenic aerosol forcing (Menary et al., 2020). A handful of CMIP6 realizations (5 of 92) yield AMOC evolution since 1900 similar to the indirect observations, implying the inferred AMOC weakening from 1950-1990 (and even
from 1930-1990) may have a significant contribution from internal (i.e., unforced) climate variability.

Consistent with the recent decreases in anthropogenic aerosol emissions, nearly all of the future emission scenarios (Shared Socio-economic Pathways, SSPs) (O'Neill et al., 2014) yield large reductions in future anthropogenic aerosol emissions, with

global sulfate emissions projected to decrease by up to 80% by 2050. Thus, anthropogenic aerosol emissions, including those around the North Atlantic, will likely continue to rapidly decline over the next few decades. Our results suggest that the continued decrease in anthropogenic aerosol emissions that accompany efforts to reduce air pollution will reinforce GHG-induced AMOC weakening over the next few decades−with the caveat that internal AMOC variability will also be important.

*Code and data availability.* The data and code that support the findings of this study are available from the corresponding author upon reasonable request. CMIP6 data can be downloaded from the Earth System Grid Federation at https://esgf-node.llnl.gov/search/cmip6/, or by using the $acccmip6$ package available at https://github.com/TaufiqHassan/acccmip6. MERRA2 data can be accessed at https://gmao.gsfc. nasa.gov/reanalysis/MERRA-2/. NCEP R1 data can be accessed at https://www.esrl.noaa.gov/psd/data/gridded/data.ncep.reanalysis.html. ERA5 data can be accessed at https://www.ecmwf.int/en/forecasts/datasets/reanalysis-datasets/era5. OAFlux data can be accessed at http://oaflux.whoi.edu/. Surface temperature observations can be accessed at https://www.esrl.noaa.gov/psd/data/gridded/data.gistemp.html for GISS; https://www.esrl.noaa.gov/psd/data/gridded/data.crutem4.html#detail for CRU; and https://www.esrl.noaa.gov/psd/data/gridded/data. noaaglobaltemp.html for NOAA. RAPID AMOC data can be accessed at https://www.rapid.ac.uk/rapidmoc/rapid_data/datadl.php. NOAA NCEI ocean heat content data can be downloaded from https://www.nodc.noaa.gov/OC5/3M_HEAT_CONTENT.

*Author contributions.* R.J.A. conceived the project, designed the study, performed analyses and wrote the paper. T. H. performed data analysis and wrote the paper. W. L. and C. R. advised on methods. All authors discussed results and contributed to the writing of the manuscript.

*Competing interests.* The authors declare that they have no competing financial interests.

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

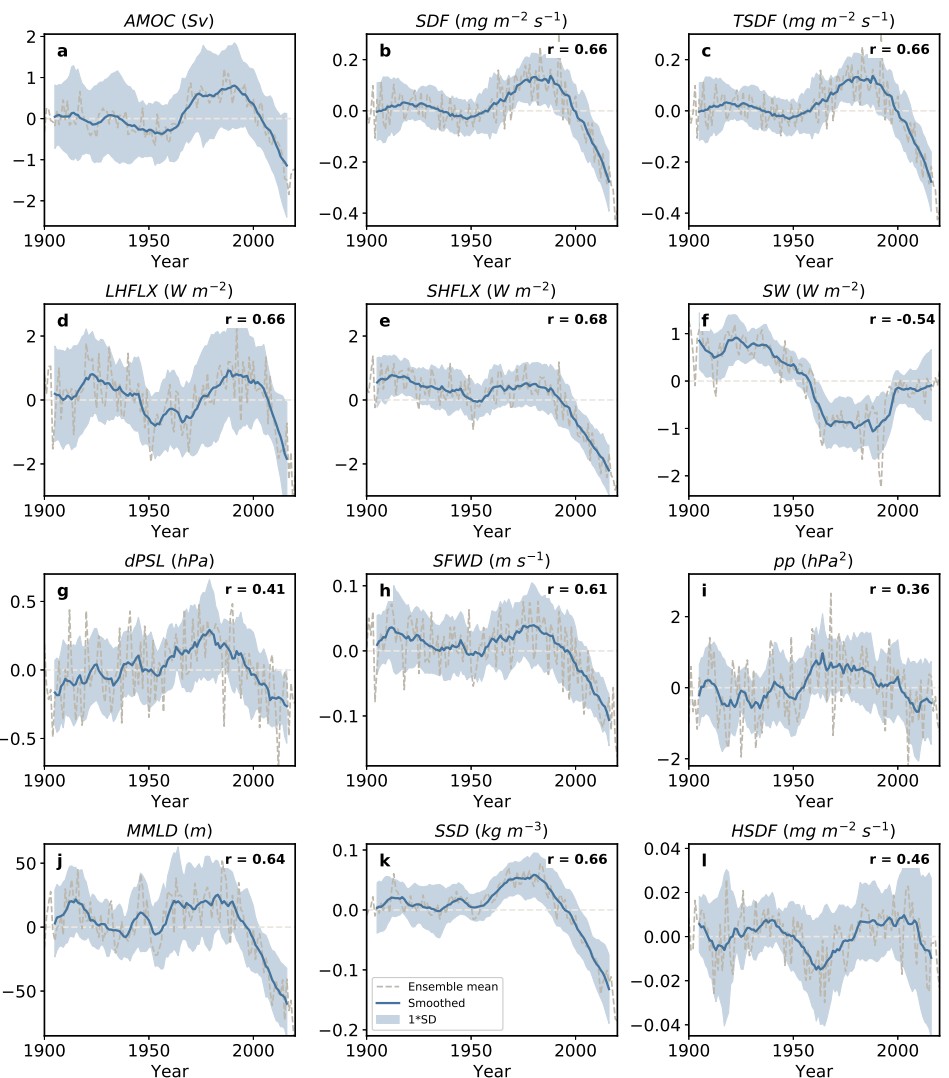

**Figure 1. 1900-2020 ensemble mean annual mean all forcing Coupled Model Intercomparison Project phase 6 normalized time series.** (a) Atlantic Meridional Overturning Circulation (AMOC) and subpolar North Atlantic (b) surface density flux (SDF); (c) thermal SDF (TSDF); (d) latent heat flux (LHFLX); (e) sensible heat flux (SHFLX); (f) net downward surface shortwave radiation (SW); (g) sea level pressure gradient (dPSL); (h) surface wind (SFWD); (i) storm track activity (pp); (j) March mixed layer depth (MMLD); (k) sea surface density (SSD); and (l) haline SDF (HSDF). The ensemble mean time series (gray dashed) is smoothed using a 10-year running mean (solid blue line). Shading shows the corresponding inter-model standard deviation. The 1950-2020 correlation coefficient (r) between the time series of each variable and the AMOC is shown in the upper right-hand side of each panel, all of which are significant at the 95% confidence level. Each model is normalized by subtracting its long-term (1900-2020) climatology. The AMOC is defined as the maximum stream function below 500 m in the Atlantic. AMOC units are Sverdrups (Sv), where 1 Sv is equal to $10^6$ $m^3$ $s^{-1}$. SDF, TSDF and HSDF units are $\frac{mg}{m^2-s}$. SHFLX, LHFLX and SW units are W $m^{-2}$. dPSL units are hPa, pp units are $hPa^2$, MMLD units are m, SSD units are kg $m^{-3}$ and SFWD units are m $s^{-1}$. The subpolar North Atlantic region is defined as 45-60°N and 0-50°W. dPSL is the Europe-subpolar North Atlantic PSL gradient defined as 30-45°N and 0-30°E minus 45-60°N and 0-50°W.

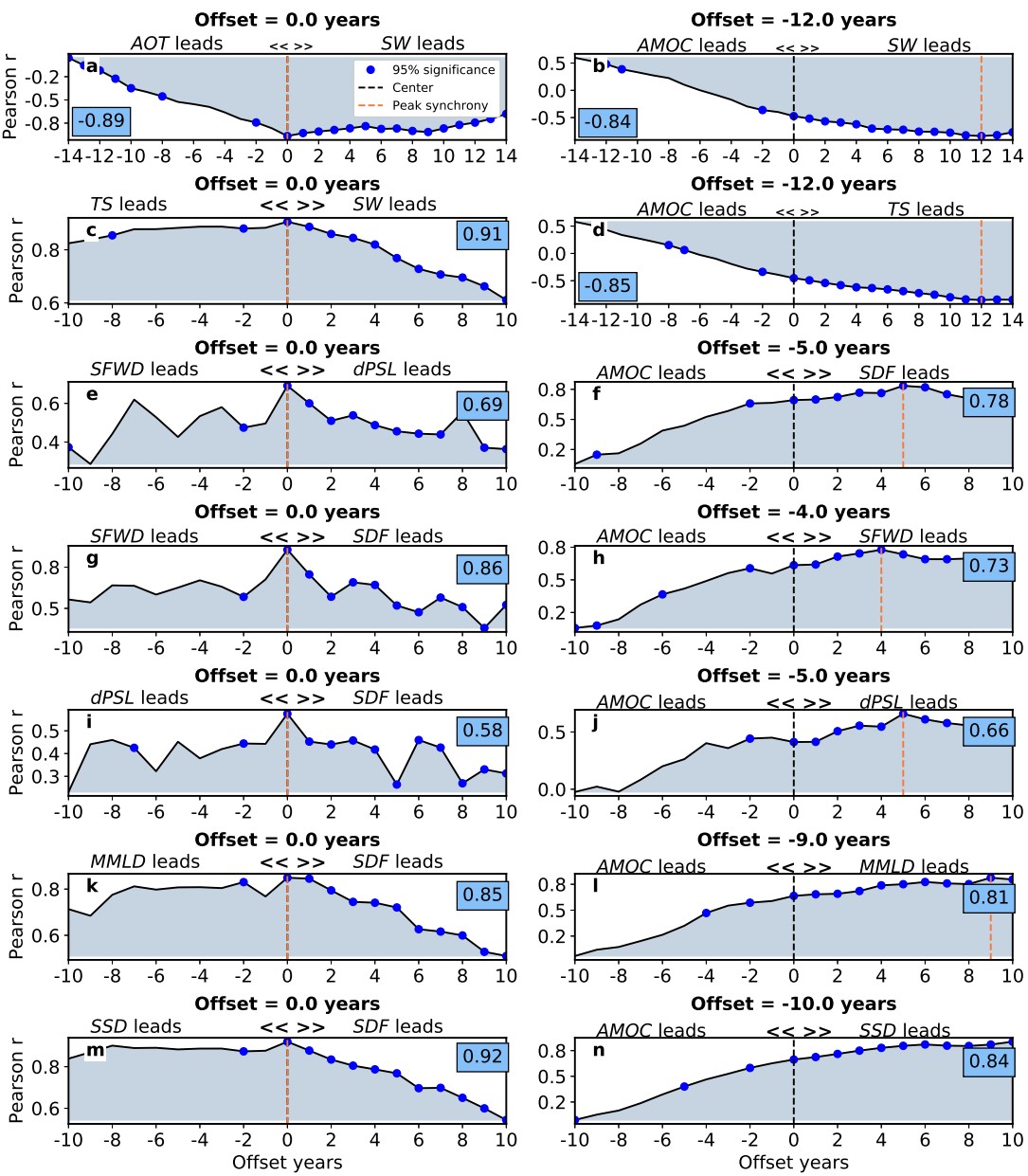

**Figure 2. 1950-2020 lead-lag AMOC correlations based on the ensemble mean annual mean all forcing Coupled Model Intercomparison Project phase 6.** (Left panels) The subpolar North Atlantic (a) aerosol optical thickness at 550 nm (AOT) versus net downward surface shortwave radiation (SW); (c) surface temperature (TS) versus SW; (e) surface wind (SFWD) versus sea level pressure gradient (dPSL); (g) SFWD versus surface density flux (SDF); (i) dPSL versus SDF; (k) March mixed layer depth (MMLD) versus SDF; and (m) sea surface density (SSD) versus SDF. (Right panels) The Atlantic Meridional Overturning Circulation (AMOC) versus the subpolar North Atlantic (b) SW; (d) TS; (f) SDF; (h) SFWD; (j) dPSL; (l) MMLD; and (n) SSD. The maximum correlation is denoted by text in the blue box. Blue filled circles denote correlations that are significant at the 95% confidence level. The corresponding offset in years is denoted by the vertical dashed red line.

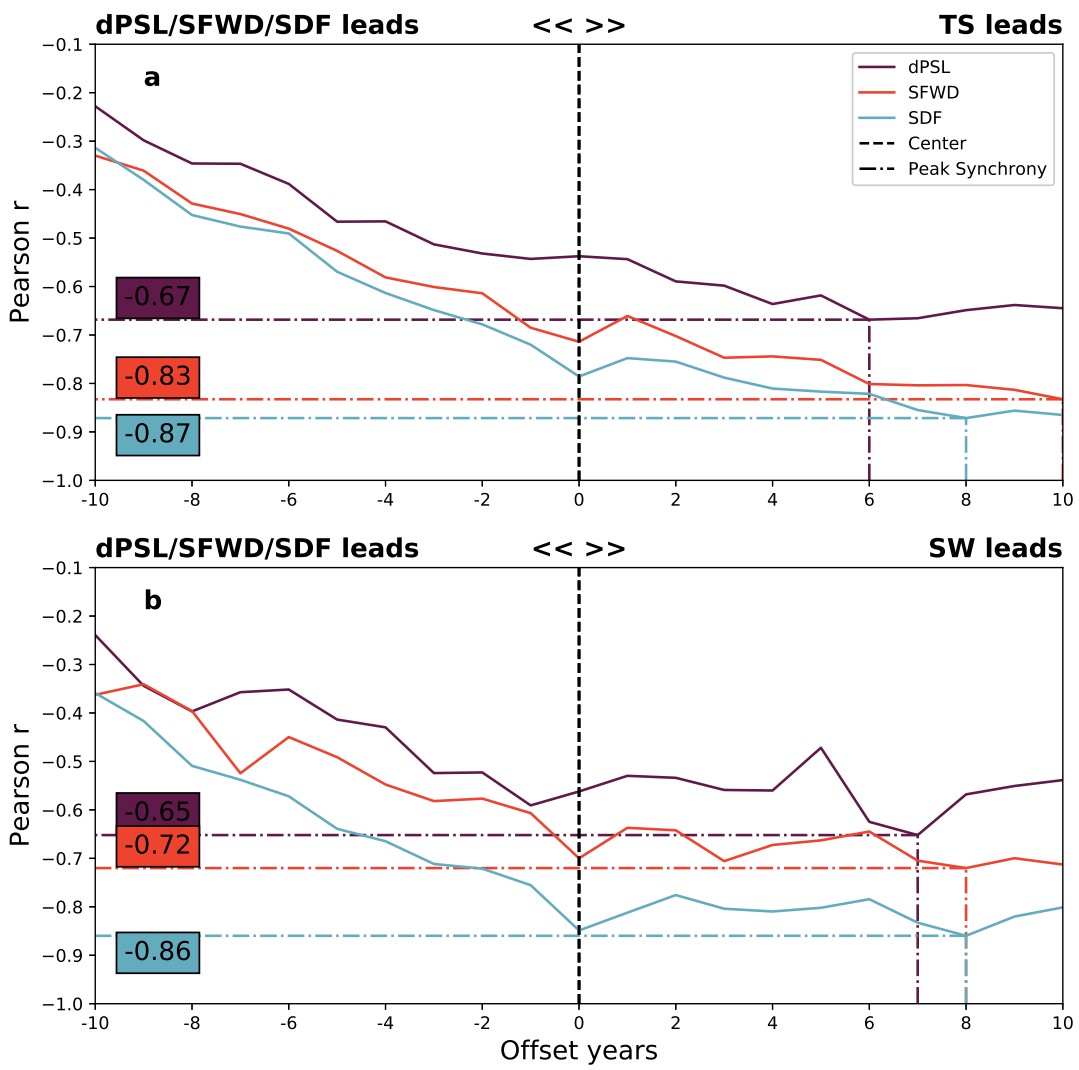

**Figure 3. 1950-2020 lead-lag atmospheric circulation correlations based on the ensemble mean annual mean all forcing Coupled Model Intercomparison Project phase 6.** The subpolar North Atlantic (a) surface temperature (TS) and (b) net downward surface shortwave radiation (SW) versus sea level pressure gradient (dPSL; purple); surface wind (SFWD; red); and surface density flux (SDF; cyan). The maximum correlations are denoted by the horizontal dashed lines and as text in the color-coded boxes, all of which are significant at the 95% confidence level. The corresponding offsets in years are denoted by the vertical dashed color-coded lines.

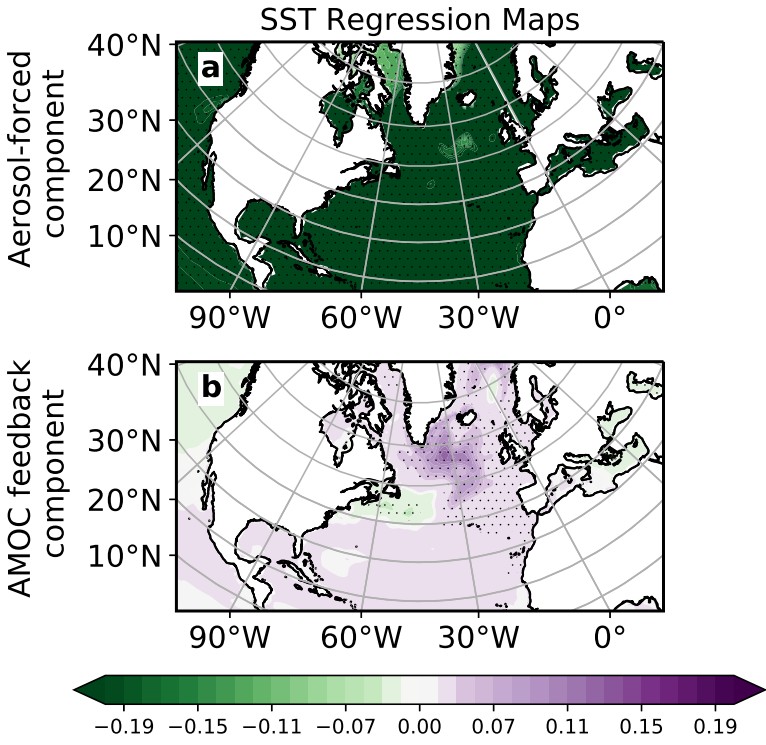

**Figure 4. 1940-2020 ensemble mean annual mean CMIP6 all forcing regression analysis.** Decomposition of sea surface temperature (SST) into (a) aerosol forced and (b) AMOC feedback components. The forced response is obtained by regressing the subpolar North Atlantic -1xSW time series (a proxy for anthropogenic aerosols) onto each field. The AMOC-related feedback is obtained by removing the variability associated with the forced response, and then regressing the AMOC time series onto this new field. The feedback field is converted to the same units as the aerosol-forced field by multiplying the feedback field by the regression slope between the AMOC and -1xSW subpolar North Atlantic time series $\frac{\delta(AMOC)}{\delta(-1 \times SW)} = 0.32 \frac{Sv}{W\ m^{-2}}$, significant at the 95% confidence level). The units for the SST regression maps are K/W m$^{-2}$. Symbols denote regression significance at the 95% confidence level.

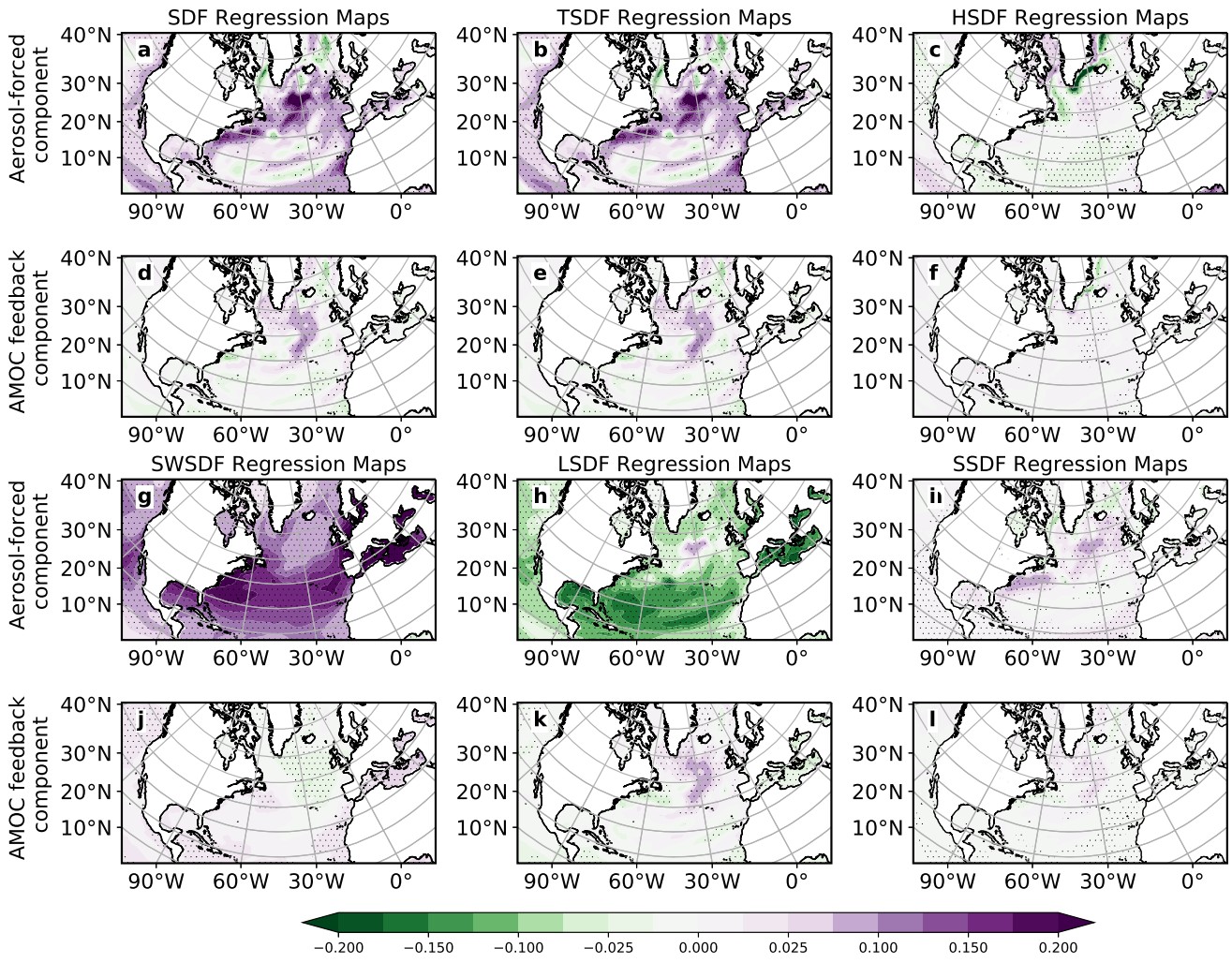

**Figure 5. 1940-2020 ensemble mean annual mean CMIP6 all forcing regression analysis.** Decomposition of (a,d) surface density flux (SDF); (b,e) thermal component of SDF (TSDF); (c,f) haline component of SDF (HSDF); (g,j) net surface shortwave radiation driven SDF (SWSDF); (h,k) latent heat flux driven SDF (LSDF); and (i,l) sensible heat flux driven SDF (SSDF) into (a-c,g-i) aerosol forced and (d-f,j-l) AMOC feedback components. The forced response is obtained by regressing the subpolar North Atlantic -1xSW time series (a proxy for anthropogenic aerosols) onto each field. The AMOC-related feedback is obtained by removing the variability associated with the forced response, and then regressing the AMOC time series onto this new field. The feedback field is converted to the same units as the aerosol-forced field by multiplying the feedback field by the regression slope between the AMOC and -1xSW subpolar North Atlantic time series $\frac{\delta(AMOC)}{\delta(-1\times SW)} = 0.32 \frac{Sv}{W\ m^{-2}}$, significant at the 95% confidence level). The units for all SDF regression maps are $mg\ m^{-2}\ s^{-1}/W\ m^{-2}$. Symbols denote regression significance at the 95% confidence level.

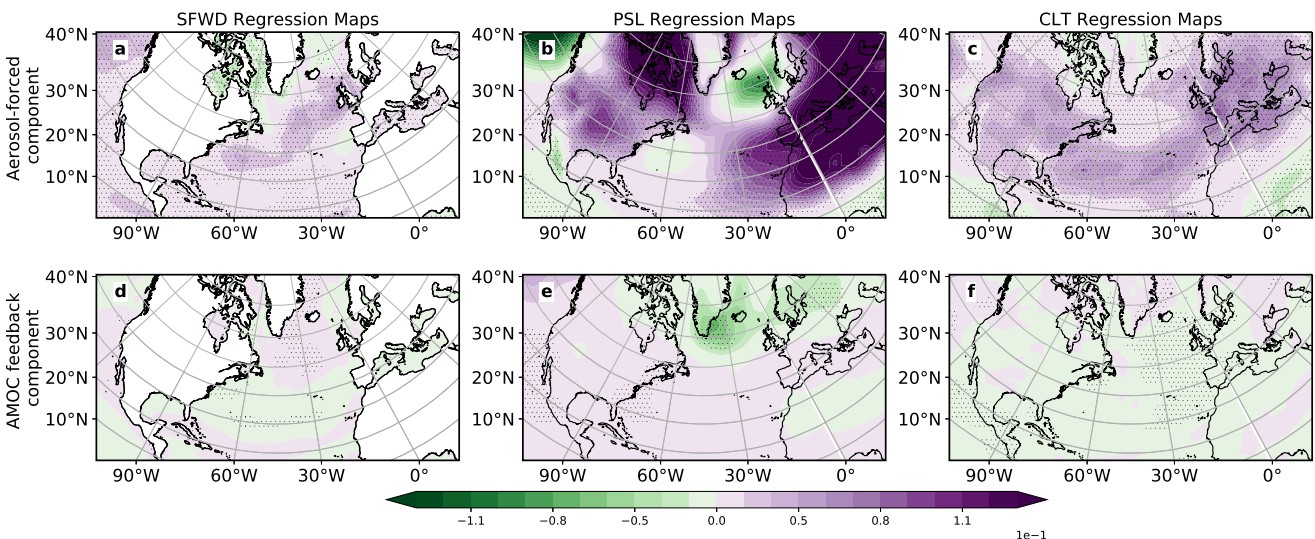

**Figure 6. 1940-2020 ensemble mean annual mean CMIP6 all forcing regression analysis.** Decomposition of (a,d) surface wind speed (SFWD); (b,e) sea level pressure (PSL); and (c,f) total cloud cover (CLT) into (top panels) aerosol forced and (bottom panels) AMOC feedback components. The forced response is obtained by regressing the subpolar North Atlantic -1xSW time series (a proxy for anthropogenic aerosols) onto each field. The AMOC-related feedback is obtained by removing the variability associated with the forced response, and then regressing the AMOC time series onto this new field. The feedback field is converted to the same units as the aerosol-forced field by multiplying the feedback field by the regression slope between the AMOC and -1xSW subpolar North Atlantic time series $\frac{\delta(AMOC)}{\delta(-1\times SW)} = 0.32$ $\frac{Sv}{W\ m^{-2}}$, significant at the 95% confidence level). The units for SFWD, PSL and CLT regression maps are m s$^{-1}$/W m$^{-2}$, hPa/W m$^{-2}$ and fraction/W m$^{-2}$ respectively. Symbols denote regression significance at the 95% confidence level.

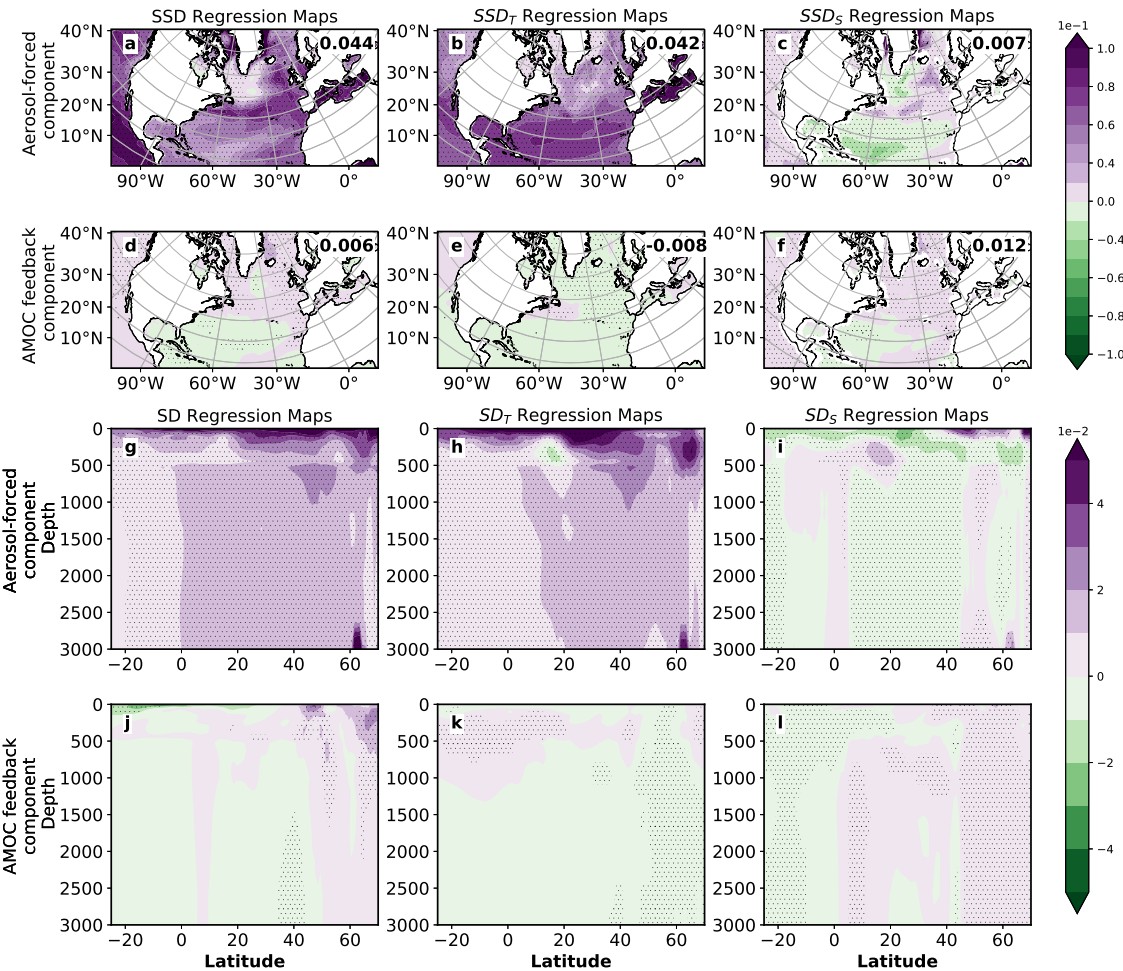

**Figure 7. 1940-2020 ensemble mean annual mean CMIP6 all forcing regression analysis.** Decomposition of (a,d) sea surface density ($SSD$); (b,e) thermal component of SSD ($SSD_T$); (c,f) haline component of SSD ($SSD_S$); (g,j) zonal mean Atlantic seawater density ($SD$); (h,k) thermal component of SD ($SD_T$); and (i,l) haline component of SD ($SD_S$) into (a-c,g-i) aerosol forced and (d-f,j-l) AMOC feedback components. The forced response is obtained by regressing the subpolar North Atlantic -1xSW time series (a proxy for anthropogenic aerosols) onto each field. The AMOC-related feedback is obtained by removing the variability associated with the forced response, and then regressing the AMOC time series onto this new field. The feedback field is converted to the same units as the aerosol-forced field by multiplying the feedback field by the regression slope between the AMOC and -1xSW subpolar North Atlantic time series $\frac{\delta(AMOC)}{\delta(-1 \times SW)} = 0.32 \frac{Sv}{W\ m^{-2}}$, significant at the 95% confidence level). Numbers in the top right of each panel show the subpolar North Atlantic averaged regression coefficients (in units of kg m$^{-3}$/W m$^{-2}$). The units for all SSD regression maps are kg m$^{-3}$/W m$^{-2}$. Symbols denote regression significance at the 95% confidence level. For the SD analysis (panels g-l), a subset of 6 CMIP6 ALL models are used, including BCC-CSM2-MR, CESM2, CNRM-CM6-1, CanESM5, HadGEM3-GC31-LL, and IPSL-CM6A-LR.

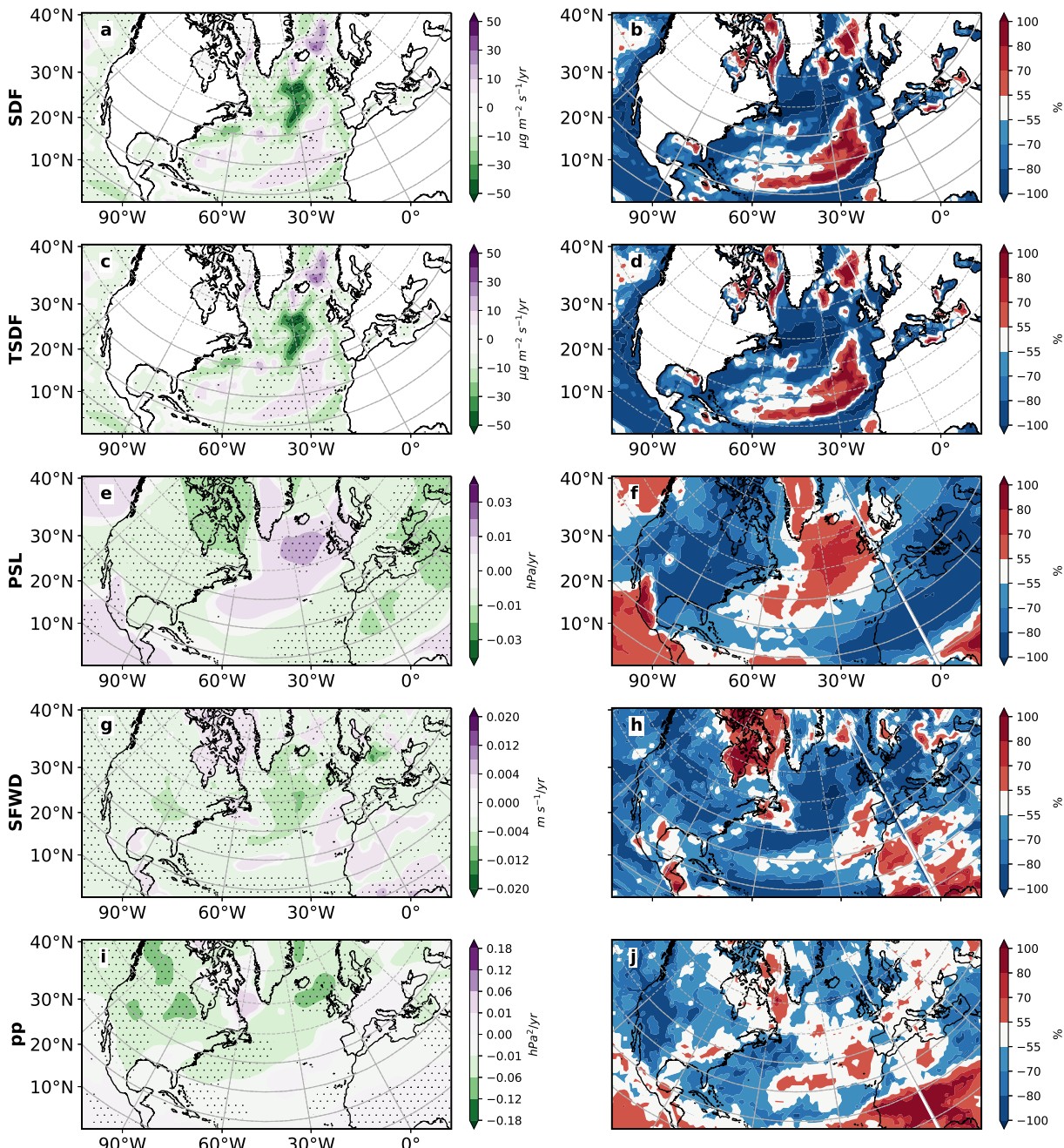

**Figure 8. 1990-2020 annual mean all forcing Coupled Model Intercomparison Project phase 6 ensemble mean trends and model agreement on the sign of the trend.** (a-b) surface density flux (SDF); (c-d) thermal SDF (TSDF); (e-f) sea level pressure (PSL); (g-h) surface winds (SFWD); and (i-j) storm track activity (pp). Left panels show the ensemble mean trend; right panels show model agreement on the sign of the trend for each model's ensemble mean. Symbols in left panels designate trend significance at the 95% confidence level based on a $t$-test. SDF and TSDF trend units are $\frac{\mu g}{m^2-s}$ year$^{-1}$. PSL, pp, and SFWD trend units are hPa year$^{-1}$, hPa$^2$ year$^{-1}$, and m s$^{-1}$ year$^{-1}$, respectively. Trend realization agreement units are %. Red (blue) colors indicate model agreement on a positive (negative) trend.

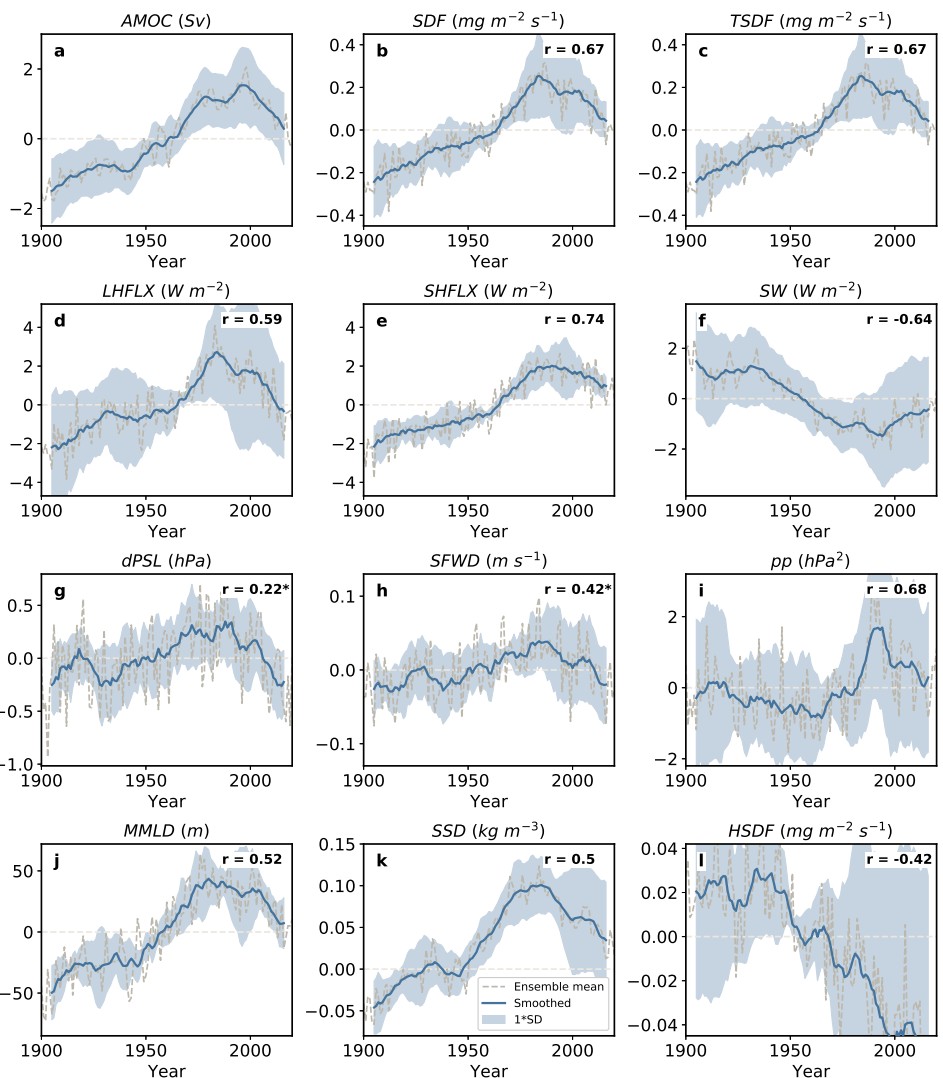

**Figure 9. 1900-2020 ensemble mean annual mean anthropogenic aerosol forcing Coupled Model Intercomparison Project phase 6 normalized time series.** (a) Atlantic Meridional Overturning Circulation (AMOC) and subpolar North Atlantic (b) surface density flux (SDF); (c) thermal SDF (TSDF); (d) latent heat flux (LHFLX); (e) sensible heat flux (SHFLX); (f) net downward surface shortwave radiation (SW); (g) sea level pressure gradient (dPSL); (h) surface wind (SFWD); (i) storm track activity (pp); (j) March mixed layer depth (MMLD); (k) sea surface density (SSD); and (l) haline SDF (HSDF). The ensemble mean time series (gray dashed) is smoothed using a 10-year running mean (solid blue line). Shading shows the corresponding inter-model standard deviation. 1950-2020 correlation coefficient against AMOC is shown at the upper right-hand side of each panel, all of which are significant at the 95% confidence level except for those correlations marked with an asterisk. Each model is normalized by subtracting its long-term (1900-2020) climatology. The AMOC is defined as the maximum stream function below 500 m in the Atlantic. AMOC units are Sverdrups (Sv), where 1 Sv is equal to $10^6$ m$^3$ s$^{-1}$. SDF, TSDF and HSDF units are $\frac{mg}{m^2-s}$. SHFLX, LHFLX and SW units are W m$^{-2}$. dPSL units are hPa, pp units are hPa$^2$, MMLD units are m, SSD units are kg m$^{-3}$ and SFWD units are m s$^{-1}$. The subpolar North Atlantic region is defined as 45-60°N and 0-50°W. dPSL is the Europe-subpolar North Atlantic PSL gradient defined as 30-45°N and 0-30°E minus 45-60°N and 0-50°W.

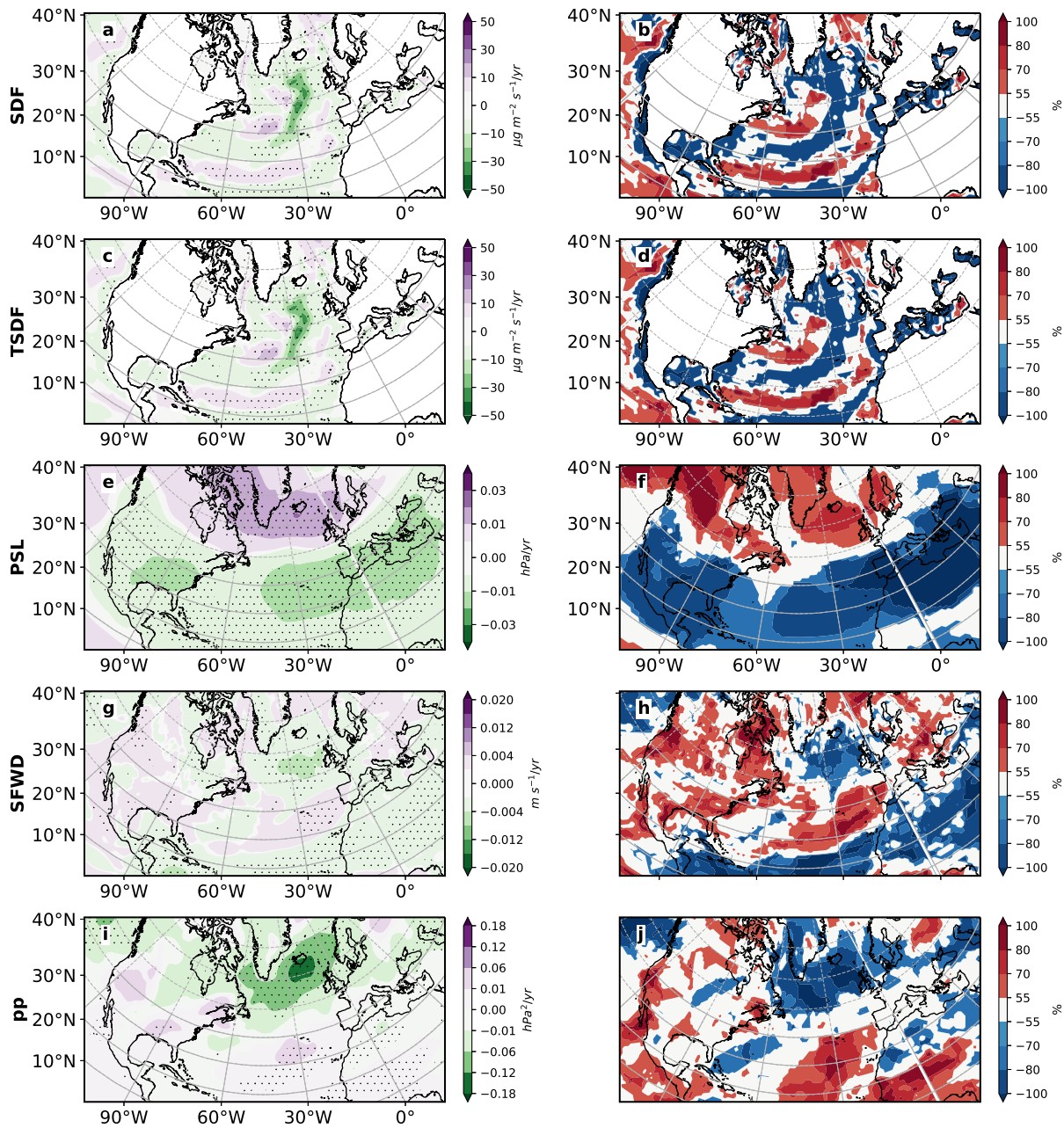

**Figure 10. 1990-2020 annual mean anthropogenic aerosol forcing Coupled Model Intercomparison Project phase 6 ensemble mean trends and model agreement on the sign of the trend.** (a-b) surface density flux (SDF); (c-d) thermal SDF (TSDF); (e-f) sea level pressure (PSL); (g-h) surface winds (SFWD); and (i-j) storm track activity (pp). Left panels show the ensemble mean trend; right panels show model agreement on the sign of the trend for each model's ensemble mean. Symbols in left panels designate trend significance at the 95% confidence level based on a $t$-test. SDF and TSDF trend units are $\frac{\mu g}{m^2 - s}$ year$^{-1}$. PSL, pp, and SFWD trend units are hPa year$^{-1}$, hPa$^2$ year$^{-1}$, and m s$^{-1}$ year$^{-1}$, respectively. Trend realization agreement units are %. Red (blue) colors indicate model agreement on a positive (negative) trend.

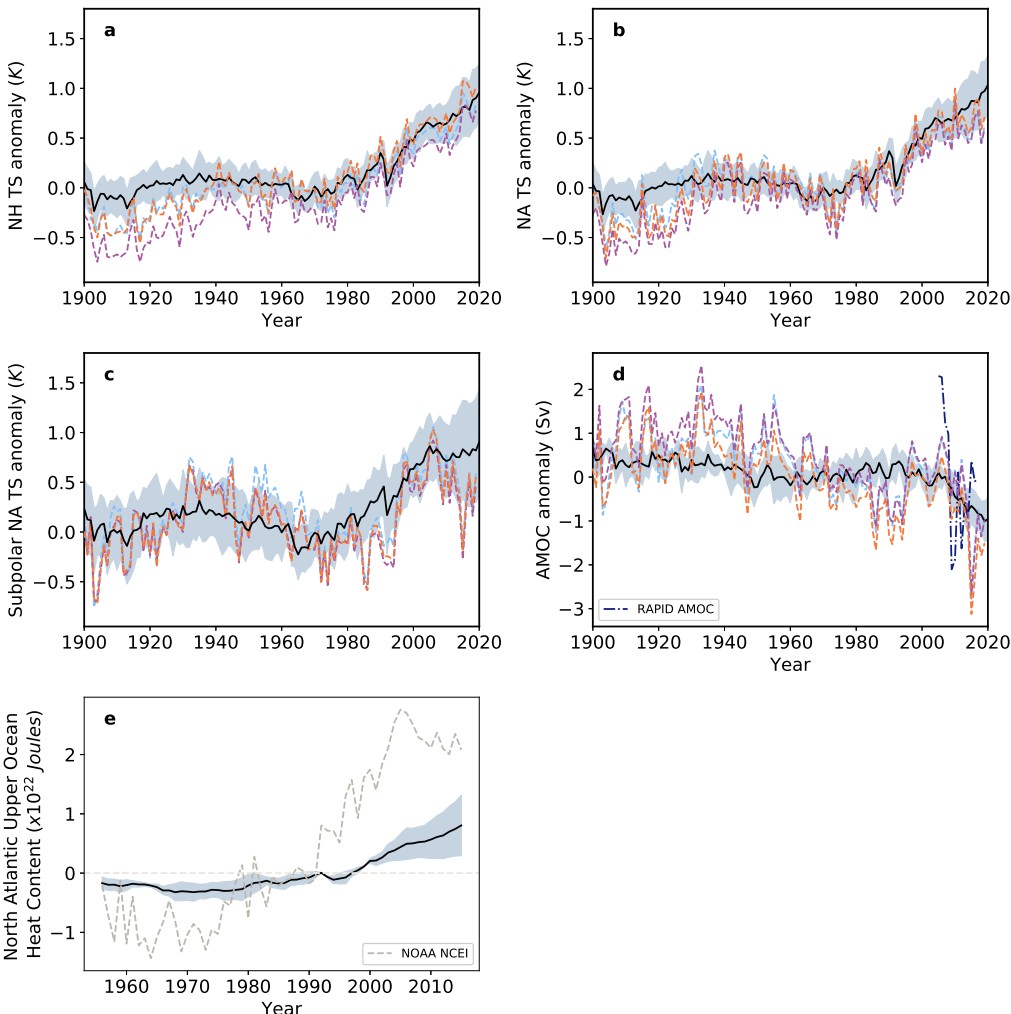

**Figure 11. Ensemble mean annual mean all forcing Coupled Model Intercomparison Project phase 6 and observed surface temperature, inferred AMOC and ocean heat content time series.** 1900-2020 (a) Northern Hemisphere (0-60°N; 0-360°E); (b) North Atlantic (0-60°N; 7.5-75°W); and (c) subpolar North Atlantic (45-60°N; 0-50°W) surface temperature (TS); (d) inferred AMOC; and (e) 1955-2014 North Atlantic upper-ocean (0-700 m) ocean heat content (OHC). The inferred AMOC is calculated in a similar fashion as in Rahmstorf et al. (2015)−as the subpolar North Atlantic (45-60°N and 0-50°W) minus the Northern Hemisphere (0-60°N and 0-360°) TS anomaly time series, scaled by 2.3 Sv K$^{-1}$. Results are shown for the CMIP6 all forcing ensemble mean (solid black) and observations (ending in 2019), including NASA GISS (GISTEMPv4; dashed light blue), NOAA (NOAAGlobalTempv4.0.1; dashed orange) and CRU (CRUTEM4; dashed green) for TS. Observed OHC (gray dashed) comes from NOAA National Centers for Environmental Information (NOAA NCEI). Also included in panel (d) is the April 2004-July 2019 directly observed AMOC from the RAPID array (dash-dot dark blue). Light blue shading shows the CMIP6 inter-model standard deviation. AMOC units are Sv; OHC units are 10$^{22}$ Joules. Inferred AMOC trends for 1950-1990 are **−0.03**, **−0.03**, **−0.03** and **0.03** Sv year$^{-1}$ for GISS, NOAA, CRU and CMIP6, respectively. The corresponding 1990-2020 inferred AMOC trends are −0.02, −0.02, −0.02 and **−0.07** Sv year$^{-1}$. The CMIP6 all forcing actual AMOC trends are **0.04** and **−0.08** Sv year$^{-1}$ for 1950-1990 and 1990-2020, respectively. Bold trends are significant at the 95% confidence level.

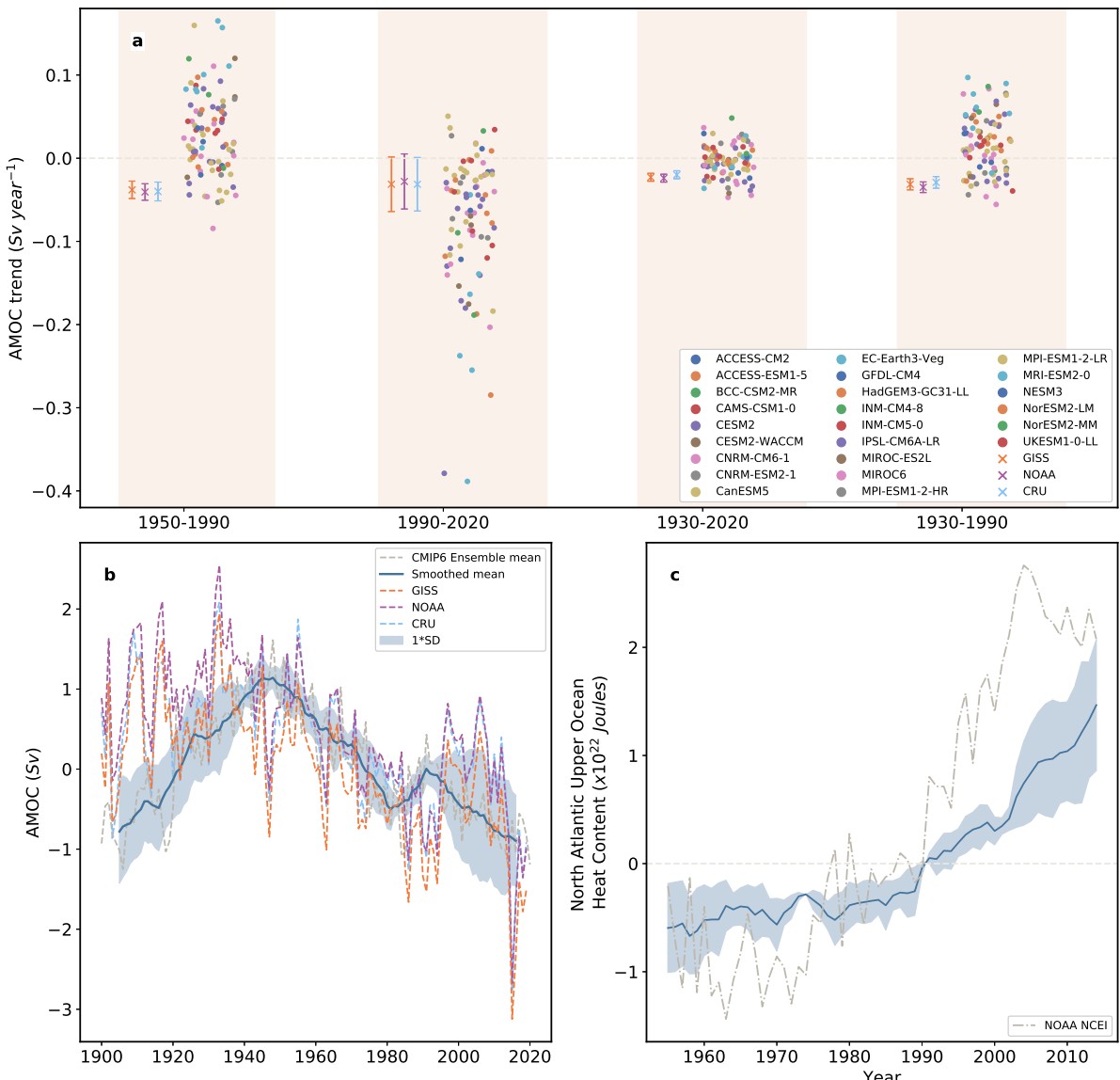

**Figure 12. Coupled Model Intercomparison Project phase 6 all forcing annual mean AMOC and OHC for individual model realizations.** (a) AMOC trends [Sv year$^{-1}$] are shown for four time periods, including 1950-1990, 1990-2020, 1930-2020, and 1930-1990. Each individual model realization is shown with a symbol. Also included (X's) are the corresponding inferred AMOC trends (ending in 2019) based on surface temperature observations. Error bars on inferred AMOC trends represent the 95% confidence interval of the trend. (b) Ensemble mean annual mean normalized AMOC [Sv] time series (gray dashed) and 10-year running mean (solid blue line) based on the two CMIP6 models (CanESM5 and IPSL-CM6A-LR; 5 realizations in total) that yield AMOC trends that fall within the observational uncertainty for all four time periods. Blue shading shows the corresponding inter-model standard deviation. Also included are the three inferred AMOC time series, based on surface temperature observations. (c) As in (b), but for the 1955-2014 North Atlantic upper-ocean (0-700 m) ocean heat content (OHC). Also included in (c) is the corresponding observed OHC (gray dashed) from NOAA National Centers for Environmental Information (NOAA NCEI) .

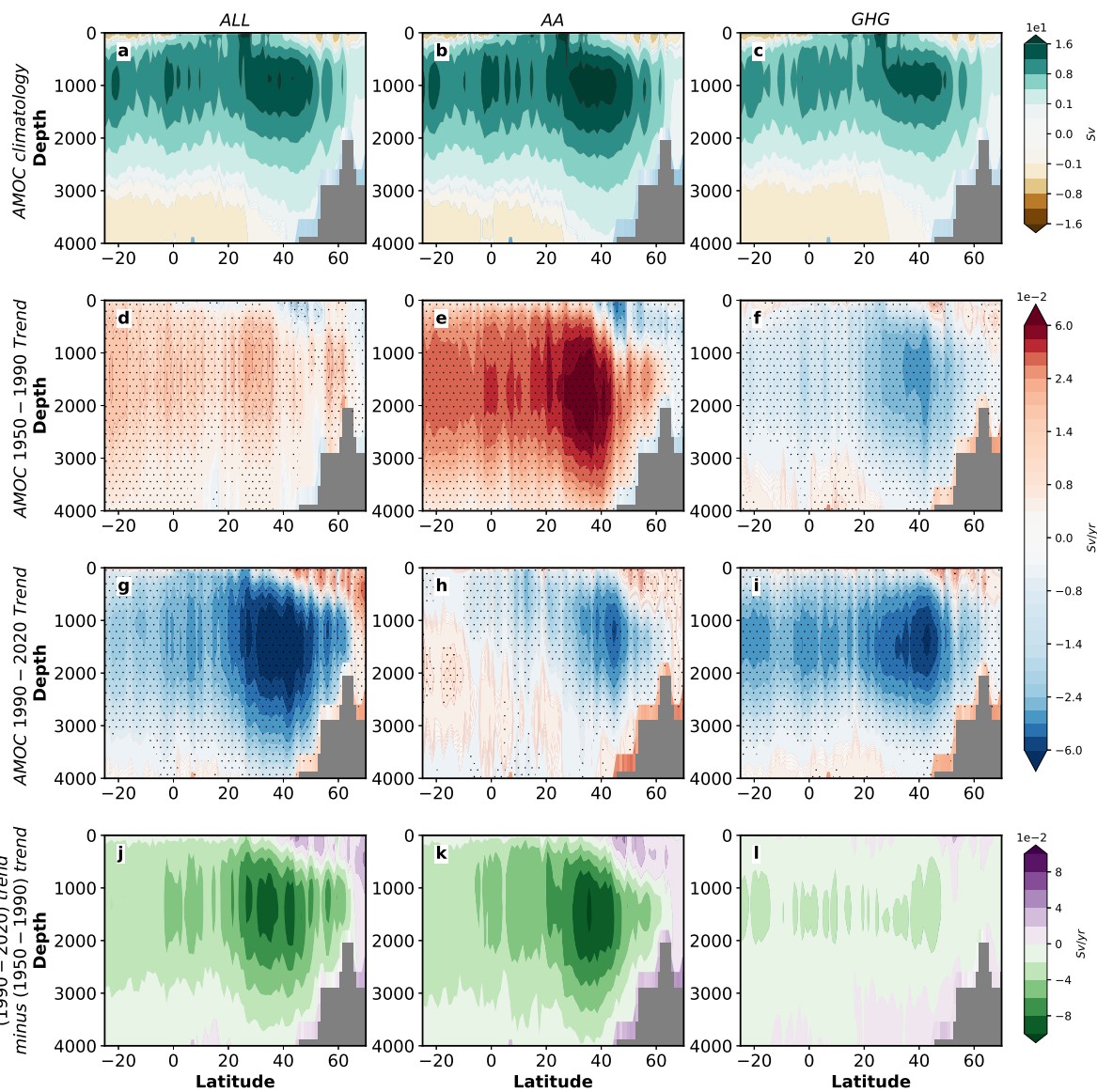

**Figure 13. 1950-2020 ensemble mean annual mean Coupled Model Intercomparison Project phase 6 Atlantic meridional stream-function in depth-latitude space.** Zonal mean (a-c) 1950-1990 climatology; (d-f) 1950-1990 trends; (g-i) 1990-2020 trends; and (j-l) trend "shift" (1990-2020 trend minus 1950-1990 trend) for (left column) all forcing; (middle column) anthropogenic aerosol forcing; and (right column) GHG forcing. Symbols designate trend significance at 95% confidence level based on a t-test. Streamfunction trend units are in Sv/year.