# Peer review of "Anthropogenic aerosol forcing of the AMOC and the associated mechanisms in CMIP6 models"

_Atmospheric Chemistry and Physics, 2020_

## Referee Comment (RC1) · Anonymous Referee #1 · 28 Oct 2020

General comments: this manuscript examines the multi-decadal variability of the AMOC and a number of atmospheric variables in the North Atlantic in the historical observations and simulations in the CMIP6 models. The main finding is the implication of anthropogenic aerosols in strengthening the AMOC during the period 1950-1990, while a reduction in aerosols afterwards lead to GHGs having a larger effect and weakening the AMOC. I think the methods and results are reasonable, and I particularly appreciated the well-balanced considerations on the roles of the anthropogenic drivers in the AMOC, the possible errors in the models, and the discussion of observational evidence, including AMOC proxies derived from other ocean variables (e.g., SST) and radiation estimates of aerosol forcing. I recommend acceptance pending minor revi-

sions.

Specific comments: - In fig. 1 please add the correlation coefficient of all the panels with panel a (AMOC) in the figures (maybe upper right?). Since in the text it is repeatedly discussed how these timeseries are related, it is important to have a quantitative reference. I am aware that this information is present in fig. 2, but it would improve the readability of the text until we get to fig. 2. - Throughout the manuscript it is unclear to me what is the role of SW radiation since it is related to a number of factors: clouds, aerosols, sea ice. Please elaborate on what you are looking at when you discuss SW, and consider using regressions (or kernels) to be more specific about the radiative contribution of each variable you are interested in examining.

Technical comments: L50: should be 'remains' L230: shold be 'significant'

---

## Referee Comment (RC2) · Anonymous Referee #2 · 7 Jan 2021

The manuscript acp-2020-769 "Anthropogenic aerosol forcing of the AMOC and the associated mechanisms in CMIP6 models" by Hassan et al. studies the AMOC variations in the 20th century CMIP6 simulations focusing on 1950 to 2020 as AMOC strengthens from 1950 to 1990 and weakens after 1990 in CMIP6 simulations. They have attributed these AMOC changes to changes in anthropogenic aerosol forcing. The main thesis of the paper is very interesting, but the authors did not really go deep enough to analyze the underlying physical processes, instead they mostly rely on the correlations. It is obvious that correlation does not mean causality. I would like the authors to do more in depth analysis on the physical processes instead of just correlation analysis before I can recommend this manuscript to be accepted for publication.

[Figure]

Comments: 1. The authors are mostly focused on the atmospheric side of changes and did not do any ocean related processes. They may look at the vertical structure change in the subpolar North Atlantic, such as an area mean vertical profile of T, S, and density. By doing so, it may get more insights on what processes cause the strengthening or weakening of the AMOC. 2. Some analysis on the mixed depth change may also helpful. Such as link the changes of mixed layer depth to the aerosol forcing and explore how the aerosol forcing can affect the deep convection in the models. 3. A comparison of the Atlantic meridional streamfunction between all forcing runs and anthropogenic aerosol runs may also help to explore the underlying physical processes.

---

## Author Comment (AC1) · 18 Feb 2021

**Response to Reviewer # 1**

We thank anonymous Reviewer # 1 for evaluating our manuscript. Below, we list our responses to each comment (in blue). We have updated Figure 1 (and others), and have added a new regression analysis, which better illustrates the role of aerosols via shortwave radiation perturbations (and the role of clouds) in driving multidecadal AMOC variability.

**Reviewer # 1**

General Comments:

This manuscript examines the multi-decadal variability of the AMOC and a number of atmospheric variables in the North Atlantic in the historical observations and simulations in the CMIP6 models. The main finding is the implication of anthropogenic aerosols in strengthening the AMOC during the period 1950-1990, while a reduction in aerosols afterwards lead to GHGs having a larger effect and weakening the AMOC. I think the methods and results are reasonable, and I particularly appreciated the well-balanced considerations on the roles of the anthropogenic drivers in the AMOC, the possible errors in the models, and the discussion of observational evidence, including AMOC proxies derived from other ocean variables (e.g., SST) and radiation estimates of aerosol forcing. I recommend acceptance pending minor revisions.

Specific Comments:

In fig. 1 please add the correlation coefficient of all the panels with panel a (AMOC) in the figures (maybe upper right?). Since in the text it is repeatedly discussed how these timeseries are related, it is important to have a quantitative reference. I am aware that this information is present in fig. 2, but it would improve the readability of the text until we get to fig. 2.

Thank you for the comment. We have added the 1950-2020 correlation coefficient (r) between the time series of each variable in Figure 1 and the AMOC. These correlation coefficients are shown in the upper right-hand side of each panel.

We have also included three more panels in Figure 1, including March mixed layer depth (MMLD), sea surface density (SSD) and the haline component of the surface density flux (HSDF). These new variables are also added to the corresponding figure that shows time series based on CMIP6 AA simulations.

Throughout the manuscript it is unclear to me what is the role of SW radiation since it is related to a number of factors: clouds, aerosols, sea ice. Please elaborate on what you are looking at when you discuss SW, and consider using regressions (or kernels) to be more specific about the radiative contribution of each variable you are interested in examining.

The multi-decadal evolution of subpolar North Atlantic net surface shortwave radiation from 1950-2020 is largely driven by changes in anthropogenic aerosol emissions. Several figures support this claim, including close correspondence between subpolar North Atlantic SW radiation ensemble mean time series in CMIP6 ALL and CMIP6 AA (the latter being driven by changes in AA only). Our lead-lag plots also show that subpolar North Atlantic SW and aerosol

optical thickness (AOT) are significantly correlated at -0.89 (no lag). Two figures in the supplement also show close correspondence between AOT and SW spatial trends.

We also note that subpolar North Atlantic SW is in phase with sea surface temperature (SST) with a high correlation of 0.91. Both SW and SST lead the AMOC by ~12 years, with a maximum correlation of -0.84 and -0.85, respectively. Thus, the lead-lag correlation analysis shows that AOT, SW and SST are highly correlated, in phase, and lead the AMOC by ~decade. This suggests AOT, via perturbations to SW and SST, is driving the AMOC changes. We have elaborated upon this in the revision.

Additional statistical analyses (via regressions) are now included to further show the role of net surface shortwave radiation (and other variables). This new analysis is in subsection 3.1.3 "Regression Decomposition into Aerosol-Forced and AMOC Feedback Components", and new figures have been added to the revision.

We decomposed the North Atlantic climate response into an anthropogenic aerosol-forced component and a subsequent AMOC-related feedback. We use the negative of net downward surface shortwave radiation (-1xSW) as a proxy for the change in anthropogenic aerosols (similar results are obtained if we use AOT). The forced response is obtained by regressing -1xSW onto different fields such as sea surface temperature (SST), surface wind speed (SFWD), sea level pressure (PSL), etc. The regression coefficients are based on linear least-squares regression analysis applied to the CMIP6 ensemble annual mean. We subsequently remove this anthropogenic aerosol related variability to isolate the AMOC related feedback, by regressing the AMOC time series onto the new field (with aerosol-related variability removed). This regression method is described in section, 2.4 "Regression Analysis".

Our new regression analysis shows that SSTs have a significant negative sensitivity (and the temperature component of sea surface density, SSD, has a significant positive sensitivity) to changes in subpolar North Atlantic -1xSW, meaning that an increase in aerosols is associated with SST cooling (and increased SSD) and vice versa (Figure R1 below). On the other hand, the SST-AMOC feedback component shows the opposite sensitivity. That is, an increase in aerosols initially cools SSTs and increases SSD, which promotes deep convection in the subpolar North Atlantic and eventually an enhanced AMOC; the enhanced AMOC (the feedback) acts to counter the cooler SSTs/enhanced SSD, due to enhanced poleward oceanic heat transport by a stronger AMOC. This is consistent with prior studies that have suggested AMOC weakening is associate with the "warming hole" in the subpolar North Atlantic.

[Figure]

**Figure R1.** *Ensemble mean annual mean CMIP6 all forcing regression analysis. Decomposition of sea surface temperature (SST) into (top panel) aerosol forced and (bottom panel) AMOC feedback components. The forced response is obtained by regressing the subpolar North Atlantic -1xSW time series (a proxy for anthropogenic aerosols) onto each field. The AMOC-related feedback is obtained by removing the variability associated with the forced response, and then regressing the AMOC time series onto this new field. The feedback field is converted to the same units as the aerosol-forced field by multiplying the feedback field by the regression slope between the AMOC and -1xSW subpolar North Atlantic time series ($\delta(AMOC)/\delta(-1xSW) = 0.32$ Sv/(W m$^{-2}$), significant at the 95% confidence level). The units for the SST regression maps are K/(W m$^{-2}$). Symbols denote regression significance at the 95% confidence level.*

We have also conducted similar regression analyses for other climate variables, including total cloud cover (CLT; Figure R2 below). The aerosol forced response show positive subpolar North Atlantic sensitivities, which is consistent with aerosols increasing cloud cover. This would act to reinforce cooling of subpolar North Atlantic SSTs, strengthening the forced component of aerosols on the AMOC. In contrast, the AMOC feedback shows opposite signed CLT sensitivities, which is likely related to the aforementioned AMOC feedback on SST and subsequent impacts on lower-tropospheric atmospheric stability. This, and additional information, is elaborated upon in the revision.

[Figure]

**Figure R2.** *Ensemble mean annual mean CMIP6 all forcing regression analysis. Decomposition of total cloud cover (CLT) into (top panel) aerosol forced and (bottom panel) AMOC feedback components. The forced response is obtained by regressing the subpolar North Atlantic -1xSW time series (a proxy for anthropogenic aerosols) onto each field. The AMOC-related feedback is obtained by removing the variability associated with the forced response, and then regressing the AMOC time series onto this new field. The feedback field is converted to the same units as the aerosol-forced field by multiplying the feedback field by the regression slope between the AMOC and -1xSW subpolar North Atlantic time series ($\delta(AMOC)/\delta(-1xSW) = 0.32$ Sv/(W m$^{-2}$), significant at the 95% confidence level). The units for the CLT regression maps are fraction/(W m$^{-2}$). Symbols denote regression significance at the 95% confidence level.*

Technical Comments:

L50: should be 'remains'

Fixed.

L230: should be 'significant'

This paragraph has been removed and replaced with a new regression analysis.

---

## Author Comment (AC2) · 18 Feb 2021

**Response to Reviewer # 2**

We thank anonymous reviewer # 2 for evaluating our manuscript. Below, we list our responses to each comment (in blue). We have added new analyses and figures related to ocean variables (e.g., mixed layer depth, sea surface density, etc.). We also include ocean subsurface analyses, including a new latitude versus depth AMOC figure. A new analysis that employs a regression-based method to further understand the mechanisms by which aerosols perturb the AMOC is also included in the revision. In particular, we decompose North Atlantic climate variables (e.g., SST, SSD) into an aerosol-forced response and a subsequent AMOC-related feedback.

**Reviewer # 2**

The manuscript acp-2020-769 "Anthropogenic aerosol forcing of the AMOC and the associated mechanisms in CMIP6 models" by Hassan et al. studies the AMOC variations in the 20th century CMIP6 simulations focusing on 1950 to 2020 as AMOC strengthens from 1950 to 1990 and weakens after 1990 in CMIP6 simulations. They have attributed these AMOC changes to changes in anthropogenic aerosol forcing. The main thesis of the paper is very interesting, but the authors did not really go deep enough to analyze the underlying physical processes, instead they mostly rely on the correlations. It is obvious that correlation does not mean causality. I would like the authors to do more in depth analysis on the physical processes instead of just correlation analysis before I can recommend this manuscript to be accepted for publication.

Comments:

1. The authors are mostly focused on the atmospheric side of changes and did not do any ocean related processes. They may look at the vertical structure change in the subpolar North Atlantic, such as an area mean vertical profile of T, S, and density. By doing so, it may get more insights on what processes cause the strengthening or weakening of the AMOC.

Thank you for the comment. We first note that we have added analyses that includes subpolar North Atlantic sea surface density (SSD), and its thermal ($SSD_T$) and haline ($SSD_S$) components. Our SSD analysis is very similar to that based on the surface density flux (SDF). For example, the SSD time series exhibits temporal evolution consistent with SDF (and the AMOC), including an increase (decrease) from ~1950-1990 (1990-2020). We have also added SSD to our lead-lag correlation analysis. SSD is in phase with SDF (with a significant correlation of 0.92 at no lag), and leads the AMOC by ~10 years (r = 0.84; significant at the 95% confidence level). Similar to SSD, $SSD_T$ also leads the AMOC, whereas $SSD_S$ lags the AMOC by ~4 years.

Additional statistical analysis (via regressions) is now included. This new analysis is in subsection 3.1.3 "Regression Decomposition into Aerosol-Forced and AMOC Feedback Components", and new figures have been added to the revision.

We decomposed the North Atlantic climate response into an anthropogenic aerosol-forced component and a subsequent AMOC-related feedback. We use the negative of net downward surface shortwave radiation (-1xSW) as a proxy for the change in anthropogenic aerosols (similar results are obtained if we use AOT). The forced response is obtained by regressing -

1xSW onto different fields such as sea surface temperature (SST), sea surface density (SSD, $SSD_T$ and $SSD_S$), and others. The regression coefficients are based on linear least-squares regression analysis applied to the CMIP6 ensemble annual mean. We subsequently remove this anthropogenic aerosol related variability to isolate the AMOC related feedback, by regressing the AMOC time series onto the new field (with aerosol-related variability removed). This regression method is described in section, 2.4 "Regression Analysis".

Figure R1 below shows the SSD regression decomposition into aerosol-forced and AMOC feedback components. As expected, a positive aerosol-forced sensitivity exists for SSD. This is largely consistent with $SSD_T$ as opposed to $SSD_S$, although $SSD_S$ also contributes near the eastern boundary of the North Atlantic. Averaged over the subpolar North Atlantic, $SSD_T$ yields an aerosol-forced sensitivity of 0.042 (kg m$^{-3}$)/(W m$^{-2}$), whereas $SSD_S$ yields a corresponding sensitivity of 0.007 (kg m$^{-3}$)/(W m$^{-2}$). The sum of these two yield 0.049 (kg m$^{-3}$)/(W m$^{-2}$), which is similar to but not exactly the same as the overall SSD sensitivity of 0.044 (kg m$^{-3}$)/(W m$^{-2}$). We note that the relative importance of salinity to the aerosol-forced SSD regression (especially along the eastern boundary of the North Atlantic) is more important than the haline component was for SDF. This difference is likely related to salt advection, which is not directly included in the SDF calculations, but is implicitly included in SSD calculations. Nonetheless, this new analysis suggests multi-decadal AMOC variability is initiated by North Atlantic aerosol perturbations to net surface shortwave radiation and surface temperature, which in turn impacts $SSD_T$.

The AMOC feedback shows similar positive sensitivities for SSD, particularly along the eastern boundary of the North Atlantic, and this is consistent with $SSD_S$. Averaged over the subpolar North Atlantic, SSD yields an AMOC feedback sensitivity of 0.006 (kg m$^{-3}$)/(W m$^{-2}$), which is entirely due to the $SSD_S$ feedback sensitivity of 0.012 (kg m$^{-3}$)/(W m$^{-2}$). The $SSD_T$ feedback sensitivity is of opposite sign, with a subpolar North Atlantic sensitivity of -0.008 (kg m$^{-3}$)/(W m$^{-2}$), implying the temperature component of SSD acts to weaken the overall SSD AMOC feedback. This is consistent with the AMOC feedback on SST (positive sensitivities; as discussed in the revision). Thus, the AMOC feedback acts to strengthen the SSD response to aerosols, and this feedback is largely due to salinity. Moreover, this salinity AMOC feedback is larger in magnitude than the aerosol-forced salinity sensitivities at 0.012 versus 0.007 (kg m$^{-3}$)/(W m$^{-2}$), respectively.

[Figure]

**Figure R1.** *Ensemble mean annual mean CMIP6 all forcing regression analysis. Decomposition of (a,d) sea surface density (SSD); (b,e) thermal component of SSD ($SSD_T$); and (c,f) haline component of SSD ($SSD_S$) into (top panels) aerosol forced and (bottom panels) AMOC feedback components. The forced response is obtained by regressing the subpolar North Atlantic -1xSW time series (a proxy for anthropogenic aerosols) onto each field. The AMOC-related feedback is obtained by removing the variability associated with the forced response, and then regressing the AMOC time series onto this new field. The feedback field is converted to the same units as the aerosol-forced field by multiplying the feedback field by the regression slope between the AMOC and -1xSW subpolar North Atlantic time series ($\delta(AMOC)/\delta(-1xSW) = 0.32$ Sv/(W $m^{-2}$), significant at the 95% confidence level). The units for all SSD regression maps are (kg $m^{-3}$)/(W $m^{-2}$). Symbols denote regression significance at the 95% confidence level. Numbers in the top right of each panel show the subpolar North Atlantic averaged regression coefficients in units of (kg $m^{-3}$)/(W $m^{-2}$).*

Moreover, a similar analysis but based on zonal mean Atlantic subsurface sea density (SD) and its thermal and haline components ($SD_T$, $SD_S$) yields similar results, as shown below:

[Figure]

**Figure R2**. *Ensemble mean annual mean CMIP6 all forcing regression analysis. Decomposition of zonal mean Atlantic (a,d) seawater density (SD); (b,e) thermal component of SD ($SD_T$); and (c,f) haline component of SD ($SD_S$) into (top panels) aerosol forced and (bottom panels) AMOC feedback components. Symbols denote regression significance at the 95% confidence level. Units are $(kg\ m^{-3})/(W\ m^{-2})$. A smaller subset of CMIP6 ALL models is used here.*

2. Some analysis on the mixed depth change may also helpful. Such as link the changes of mixed layer depth to the aerosol forcing and explore how the aerosol forcing can affect the deep convection in the models.

We have added a mixed layer depth analysis to our revised manuscript. We use March mixed layer depth (MMLD) to investigate North Atlantic deep convection, which is associated with deep water formation and the strength of the AMOC.

We have added subpolar North Atlantic MMLD in Figure 1j. The MMLD temporal evolution is similar to multi-decadal variations in SSD, SDF and AMOC, including an increase (decrease) from ~1950-1990 (1990-2020). We have also added MMLD to our lead-lag correlations analysis (Figure 2k-l). While March mixed layer depth is in phase with surface density flux (with a maximum correlation of 0.85), it leads AMOC by 9 years with a maximum correlation of 0.81.

Supplementary Figure 4 is also added, which shows the spatial trends and temporal evolution of MMLD in all forcing and anthropogenic aerosol forcing scenario. For both cases, 1990-2020 (1950-1990) CMIP6 ensemble mean annual mean show a significant decrease (increase) in wintertime deep convection in the subpolar North Atlantic.

Applying our regression analysis to MMLD shows that the aerosol-forced March mixed layer depth exhibits significant positive sensitivities in the subpolar North Atlantic (implying enhanced deep convection in response to aerosol forcing), and the corresponding MMLD-AMOC feedback also exhibits positive (but somewhat weaker) sensitivities. This latter result again implies that the AMOC induces changes that positively feedback onto the AMOC (e.g., the aforementioned salinity contribution to $SSD_S$).

3. A comparison of the Atlantic meridional streamfunction between all forcing runs and anthropogenic aerosol runs may also help to explore the underlying physical processes.

We have added a figure to the revision (Figure R3 below) that shows the CMIP6 Atlantic meridional streamfunction in depth-latitude space, which is calculated from zonally integrated meridional velocity field. We use a common set of models from the CMIP6 ALL, AA and GHG forcing experiments.

The 1990-2020 CMIP6 AMOC weakening is significant throughout most of the North Atlantic in all three forcing scenarios–ALL, AA and GHG–with GHG weakening larger than that due to AA. In contrast, the 1950-1990 time period features CMIP6 ALL and AA strengthening that is again significant throughout most of the North Atlantic; CMIP6 GHG forcing yields the opposite response (and weaker than the CMIP6 AA strengthening). Thus, the 1950-1990 AMOC strengthening in CMIP6 ALL is entirely dominated by AA, with GHGs acting to mute this

strengthening. The 1990-2020 AMOC weakening in CMIP6 ALL is due to both GHGs and AAs, with GHGs driving a larger response. To measure the overall 1950-2020 impact of AA versus GHGs on the AMOC, we calculate the difference of the trends (1990-2020 minus 1950-1990). Figure R3 below shows that this trend "shift" is largely due to aerosols, as opposed to GHGs.

[Figure]

**Figure R3**. *1950-2020 ensemble mean annual mean Coupled Model Intercomparison Project phase 6 Atlantic meridional streamfunction in depth-latitude space. Zonal mean (a-c) 1950-1990 climatology; (d-f) 1950-1990 trends; (g-i) 1990-2020 trends; and (j-l) trend "shift" (1990-2020 trend minus 1950-1990 trend) for (left column) all forcing; (middle column) anthropogenic aerosol forcing; and (right column) GHG forcing. Symbols designate trend significance at 95%*